# BYZANTINE-RESILIENT NON-CONVEX STOCHASTIC GRADIENT DESCENT*

**Zeyuan Allen-Zhu**,[†] **Faeze Ebrahimian**,[‡] **Jerry Li**,[§] **Dan Alistarh**[¶]

## ABSTRACT

We study adversary-resilient stochastic distributed optimization, in which $m$ machines can independently compute stochastic gradients, and cooperate to jointly optimize over their local objective functions. However, an $\alpha$-fraction of the machines are *Byzantine*, in that they may behave in arbitrary, adversarial ways. We consider a variant of this procedure in the challenging *non-convex* case. Our main result is a new algorithm SafeguardSGD which can provably escape saddle points and find approximate local minima of the non-convex objective. The algorithm is based on a new concentration filtering technique, and its sample and time complexity bounds match the best known theoretical bounds in the stochastic, distributed setting when no Byzantine machines are present.

Our algorithm is very practical: it improves upon the performance of all prior methods when training deep neural networks, it is relatively lightweight, and it is the first method to withstand two recently-proposed Byzantine attacks.

## 1 INTRODUCTION

Motivated by the pervasiveness of large-scale distributed machine learning, there has recently been significant interest in providing distributed optimization algorithms with strong *fault-tolerance* guarantees. In this context, the strongest, most stringent fault model is that of *Byzantine faults* (Lamport et al., 1982): given $m$ machines, each having access to private data, at most an $\alpha$ fraction of the machines can behave in arbitrary, possibly adversarial ways, with the goal of breaking or slowing down the algorithm. Although extremely harsh, this fault model is the "gold standard" in distributed computing (Lynch, 1996; Lamport et al., 1982; Castro et al., 1999), as algorithms proven to be correct in this setting are guaranteed to converge under arbitrary system behaviour.

A setting of particular interest in this context has been that of distributed *stochastic* optimization. Here, the task is to minimize some stochastic function $f(x) = \mathbb{E}_{s \sim \mathcal{D}}[f_s(x)]$ over a distribution $\mathcal{D}$, where $f_s(\cdot)$ can be viewed as the loss function for sample $s \sim \mathcal{D}$. We assume there are $m$ machines (workers) and an honest master, and $\alpha < 1/2$ fraction of the workers may be Byzantine. In each iteration $t$, each worker has access to a version of the global iterate $x_t$, which is maintained by the master. The worker can independently sample $s \sim \mathcal{D}$, compute $\nabla f_s(x_t)$, and then synchronously send this stochastic gradient to the master. The master aggregates the workers' messages, and sends an updated iterate $x_{t+1}$ to all the workers. Eventually, the master has to output an approximate minimizer of $f$. Clearly, the above description only applies to *honest* workers; Byzantine workers may deviate arbitrarily and return adversarial "gradient" vectors to the master in every iteration.

This distributed framework is quite general and well studied. One of the first references in this setting studied distributed PCA and regression (Feng et al., 2014). Other early approaches (Blanchard et al., 2017; Chen et al., 2017; Su & Vaidya, 2016a;b; Xie et al., 2018a) relied on defining generalizations of the geometric median. These approaches can withstand up to half of the nodes being malicious, but can have relatively high local computational cost $\Omega(m^2 d)$ (Blanchard et al., 2017; Chen et al., 2017), where $m$ is the number of nodes and $d$ is the problem dimension, and usually have sub-optimal sample and iteration complexities.

Follow-up work resolved this last issue when the objective $f(\cdot)$ is *convex*, leading to tight sample

---

*The full and future editions of this paper can be found on `https://arxiv.org/abs/2012.14368`.

†Microsoft Research Redmond, `zeyuan@csail.mit.edu`

‡University of Waterloo, `faezeeb75@gmail.com`

§Microsoft Research Redmond, `jerrl@microsoft.com`

¶IST Austria, `dan.alistarh@ist.ac.at`

complexity bounds. Specifically, Yin et al. (2018) provided bounds for *gradient descent*-type algorithms, and showed that the bounds are tight when the dimension is constant. Alistarh et al. (2018) provided a stochastic gradient descent (SGD) type algorithm and showed that its sample and time complexities are asymptotically optimal even when the dimension is large.

**Non-convex Byzantine-resilient stochastic optimization.** In this paper, we focus on the more challenging non-convex setting, and shoot for the strong goal of finding *approximate local minima* (a.k.a. second-order critical points). In a nutshell, our main result is the following. Fix $d$ to denote the dimension, and let the objective $f : \mathbb{R}^d \to \mathbb{R}$ be Lipschitz smooth and second-order smooth. We have $m$ worker machines, each having access to unbiased, bounded estimators of the gradient of $f$. Given an initial point $x_0$, the $\texttt{SafeguardSGD}$ algorithm ensures that, even if at most $\alpha < 1/2$ fraction of the machines are Byzantine, after

$$T = \widetilde{O}\left(\left(\alpha^2 + \tfrac{1}{m}\right)\tfrac{d(f(x_0) - \min f(x))}{\varepsilon^4}\right) \qquad \text{parallel iterations,}$$

for at least a constant fraction of the indices $t \in [T]$, the following hold:

$$\|\nabla f(x_t)\| \le \varepsilon \quad \text{and} \quad \nabla^2 f(x_t) \succeq -\sqrt{\varepsilon}\mathbf{I}.$$

If the goal is simply $\|\nabla f(x_t)\| \le \varepsilon$, then $T = \widetilde{O}\left(\left(\alpha^2 + \tfrac{1}{m}\right)\tfrac{(f(x_0) - \min f(x))}{\varepsilon^4}\right)$ iterations suffice. Here, the $\widetilde{O}$ notation serves to hide logarithmic factors for readability. We spell out these factors in the detailed analysis.

- When $\alpha < 1/\sqrt{m}$, our sample complexity ($= mT$) matches the best known result in the non-Byzantine case (Jin et al., 2019) without additional assumptions, and enjoys linear parallel speed-up: with $m$ workers of which $< \sqrt{m}$ are Byzantine, the parallel speedup is $\widetilde{\Omega}(m)$.[1]

- For $\alpha \in [1/\sqrt{m}, 1/2)$, our parallel time complexity is $\widetilde{O}(\alpha^2)$ times that needed when no parallelism is used. This still gives parallel speedup. This $\alpha^2$ factor appears in convex Byzantine distributed optimization, where it is tight (Yin et al., 2018; Alistarh et al., 2018).

- The Lipschitz and second-order smoothness assumptions are the minimal assumptions needed to derive convergence rates for finding second-order critical points (Jin et al., 2019).

**Comparison with prior bounds.** The closest known bounds are by Yin et al. (2019), who derived three gradient descent-type of algorithms (based on median, mean, and iterative filtering) to find a weaker type of approximate local minima. Since it relies on full gradients, their algorithm is arguably less practical, and their time complexities are generally higher than ours (see Section 2.1).

Other prior works consider a weaker goal: to find approximate stationary points $\|\nabla f(x)\| \le \varepsilon$ only: Bulusu et al. (2020) additionally assumed there is a guaranteed good (i.e. non-Byzantine) worker known by the master, Xie et al. (2018b) gave a practical algorithm when the Byzantine attackers have no information about the loss function or its gradient, Yang et al. (2019); Xie et al. (2018a); Blanchard et al. (2017) derived eventual convergence without an explicit complexity bound, and the non-convex result obtained in Yin et al. (2018) is subsumed by Yin et al. (2019), discussed above.

**Our algorithm and techniques.** The structure of our algorithm is deceptively simple. The master node keeps track of the *sum of gradients* produced by *each* worker across time. It labels *(allegedly) good* workers as those whose sum of gradients "concentrate" well with respect to a surrogate of the median vector, and labels *bad* workers otherwise. Once a worker is labelled bad, it is removed from consideration forever. The master then performs the *vanilla SGD*, by moving in the negative direction of the average gradients produced by those workers currently labelled as good.

We call our algorithm $\texttt{SafeguardSGD}$, since it behaves like having a safe guard to filter away bad workers. Its processing overhead at the master is $O(md)$, negligible compared to standard SGD.

As the astute reader may have guessed, the key non-trivial technical ingredient is to identify the right quantity to check for concentration, and make it compatible with the task of non-convex optimization. In particular, we manage to construct such quantities so that (1) good non-Byzantine workers never get mislabelled as bad ones; (2) Byzantine workers may be labelled as good ones (which is inevitable) but when they do, the convergence rates are not impacted significantly; and (3) the notion does not require additional assumptions or running time overhead.

The idea of using concentration (for each worker across time) to filter out Byzantine machines

---

[1]By *parallel speedup* we mean the reduction in wall-clock time due to sampling gradients in parallel among the $m$ nodes. In each time step, the algorithm generates $m$ new gradients, although some may be corrupted.

traces back to the convex setting (Alistarh et al., 2018). However, the quantities used in (Alistarh et al., 2018) to check for concentration are necessarily different from this paper, and our analysis is completely new, as deriving non-convex rates is known to be much more delicate and challenging. Recently, Bulusu et al. (2020) used similar concentration filters to Alistarh et al. (2018) in the non-convex setting, but under stronger assumptions, and for the simpler task of finding stationary points.

Many other algorithms *do not* rely on concentration filters. In each iteration, they ask each worker to compute *a batch* of stochastic gradients, and then use coordinate-wise median or mean over the batch average (e.g. Yin et al. (2018; 2019); Yang et al. (2019)) or iterative filtering (e.g. Su & Xu (2018); Yin et al. (2019)) by the master to derive a "robust mean." These works fundamentally rely on each iteration to calculate an *almost precise* full gradient, so that they can apply a surrogate of *full gradient descent*. Such algorithms can introduce higher sample and time complexities (see Section 2), are less practical than *stochastic gradient* schemes, require additional restrictions on the resilience factor $\alpha$, e.g. $\alpha < 1/4$ (Su & Xu, 2018), and, critically, have been shown to be vulnerable to recent attacks (Baruch et al., 2019; Xie et al., 2020).

**Attack resilience and experimental validation.** There is a growing literature on customized attacks against Byzantine-resilient algorithms, showing that many defenses can be entirely circumvented in real-world scenarios (Baruch et al., 2019; Xie et al., 2020). Our algorithm is provably correct against these attacks, a fact we also validate experimentally. We implemented `SafeguardSGD` to examine its practical performance against a range of prior works (Xie et al., 2018b; Blanchard et al., 2017; Chen et al., 2017; Yin et al., 2018; 2019), and against recent attacks on the distributed task of *training deep neural networks*. Our experiments show that `SafeguardSGD` generally outperforms previous methods in convergence speed and final accuracy, sometimes by a wide accuracy margin. This is true not only against known Byzantine attacks, but also against attack variants we fine-crafted to specifically slow down our algorithm, and against transient node failures.

## 2 STATEMENT OF OUR THEORETICAL RESULT

We denote by $\|\cdot\|$ the Euclidean norm and $[n] := \{1, 2, \ldots, n\}$. Given symmetric matrices $\mathbf{A}, \mathbf{B}$, we let $\|\mathbf{A}\|_2$ denote the spectral norm of $\mathbf{A}$. We use $\succeq$ to denote Loewner ordering, i.e. $\mathbf{A} \succeq \mathbf{B}$ if $\mathbf{A} - \mathbf{B}$ is positive semi-definite. We denote by $\lambda_{\min}(\mathbf{A})$ the minimum eigenvalue of matrix $\mathbf{A}$.

We consider arbitrary $d$-dimensional non-convex functions $f \colon \mathbb{R}^d \to \mathbb{R}$ satisfying the following:

- $f(x)$ is $L$-Lipschitz smooth: meaning $\|\nabla f(x) - \nabla f(y)\| \leq L\|x - y\|$ for any $x, y \in \mathbb{R}^d$;
- $f(x)$ is $L_2$-second-order smooth: $\|\nabla^2 f(x) - \nabla^2 f(y)\|_2 \leq L_2 \cdot \|x - y\|$ for any $x, y \in \mathbb{R}^d$;

For notational simplicity of the proofs, we assume $L = L_2 = \mathcal{V} = 1$.[2] Note that we have also assumed the domain of $f$ is the entire space $\mathbb{R}^d$. If instead there is a compact domain $\mathcal{X} \subset \mathbb{R}^d$, then one can use projected SGD and re-derive similar results of this paper. We choose to present our result in the simplest setting to convey our main ideas.

**Byzantine non-convex stochastic distributed optimization.** We let $m$ be the number of worker machines and assume at most an $\alpha$ fraction of them are Byzantine for $\alpha \in \left[0, \frac{1}{2}\right)$. We denote by good $\subseteq [m]$ the set of good (i.e. non-Byzantine) machines, and the algorithm does not know good.

**Assumption 2.1.** *In each iteration $t$, the algorithm (on the master) is allowed to specify a point $x_t$ and query $m$ machines. Each machine $i \in [m]$ gives back a vector $\nabla_{t,i} \in \mathbb{R}^d$ satisfying*

- *If $i \in$ good, the stochastic gradient $\nabla_{t,i}$ satisfies $\mathbb{E}[\nabla_{t,i}] = \nabla f(x_t)$ and $\|\nabla f(x_t) - \nabla_{t,i}\| \leq \mathcal{V}$.[3]*
- *If $i \in [m] \setminus$ good, then $\nabla_{t,i}$ can be arbitrary (w.l.o.g. we assume $\|\nabla f(x_t) - \nabla_{t,i}\| \leq \mathcal{V}$).[4]*

*Remark* 2.2. For each $t$ and $i \notin$ good, the vector $\nabla_{t,i}$ can be adversarially chosen and may depend

---

[2]In the literature of convergence analysis for non-convex optimization, the final complexity bounds *naturally* and *polynomially* depend on these parameters $L, L_2, \mathcal{V}$, and the way the dependence goes is typically unique (Allen-Zhu, 2018a;b; Fang et al., 2018; Jin et al., 2019). This is why it suffices to ignore their appearance and only compare the polynomial dependence on $\varepsilon$ and $d$.

[3]One can instead assume $\Pr[\|\nabla f(x_t) - \nabla_{t,i}\| > t] \leq 2\exp(-t^2/2\mathcal{V}^2)$ and the results of this paper continue to hold up to logarithmic factors. To present the simplest theory, we do not include that version in this paper. We refer interested readers to Jin et al. (2019) for how to deal with such probabilistic assumption (when there is no Byzantine worker).

[4]This requirement $\|\nabla f(x_t) - \nabla_{t,i}\| \leq \mathcal{V}$ is "without loss of generality" because it is trivial for the algorithm to catch bad machines if they output $\nabla_{t,i}$ more than $2\mathcal{V}$ away from the majorities.

---

**Algorithm 1** `SafeguardSGD`: perturbed SGD with double safe guard

---

**Input:** point $x_0 \in \mathbb{R}^d$, rate $\eta > 0$, lengths $T \geq T_1 \geq T_0 \geq 1$, threshold $\mathfrak{T}_1 > \mathfrak{T}_0 > 0$;
1: $\mathsf{good}_0 \leftarrow [m]$;
2: **for** $t \leftarrow 0$ **to** $T - 1$ **do**
3:      $last_1 \leftarrow \max\{t_1 \in [t] \colon t_1 \text{ is a multiple of } T_1\}$;
4:      $last_0 \leftarrow \max\{t_0 \in [t] \colon t_0 \text{ is a multiple of } T_0\}$
5:      **for each** $i \in \mathsf{good}_t$ **do**
6:          receive $\nabla_{t,i} \in \mathbb{R}^d$ from machine $i$;
7:          $A_i \leftarrow \sum_{k=last_1}^{t} \frac{\nabla_{k,i}}{|\mathsf{good}_k|}$ and $B_i \leftarrow \sum_{k=last_0}^{t} \frac{\nabla_{k,i}}{|\mathsf{good}_k|}$;
8:      $A_{\mathsf{med}} \leftarrow A_i$ where $i \in \mathsf{good}_t$ is any machine s.t. $\left|\{j \in \mathsf{good}_t \colon \|A_j - A_i\| \leq \mathfrak{T}_1\}\right| > m/2$.
9:      $B_{\mathsf{med}} \leftarrow B_i$ where $i \in \mathsf{good}_t$ is any machine s.t. $\left|\{j \in \mathsf{good}_t \colon \|B_j - B_i\| \leq \mathfrak{T}_0\}\right| > m/2$.
10:     $\mathsf{good}_{t+1} \leftarrow \left\{i \in \mathsf{good}_t \colon \|A_i - A_{\mathsf{med}}\| \leq 2\mathfrak{T}_1 \bigwedge \|B_i - B_{\mathsf{med}}\| \leq 2\mathfrak{T}_0\right\}$;
11:     $x_{t+1} = x_t - \eta\left(\xi_t + \frac{1}{|\mathsf{good}_t|}\sum_{i \in \mathsf{good}_t} \nabla_{t,i}\right)$;         $\diamond$ *Gaussian noise* $\xi_t \sim \mathcal{N}(0, \nu^2 \mathbf{I})$

---

on $\{\nabla_{t',i}\}_{t' \leq t, i \in [m]}$. In particular, the Byzantine machines can even collude during an iteration.

### 2.1 OUR ALGORITHM AND THEOREM

Our algorithm is based on arguably the simplest possible method for achieving this goal, *(perturbed) stochastic gradient descent (SGD)* (Ge et al., 2015). Our techniques more broadly apply to more complicated methods (e.g. at least to Allen-Zhu (2018a;b)), but we choose to analyze the simplest variant of SGD, since it is the most widely applied method in modern non-convex machine learning.

As illustrated in Algorithm 1, in each iteration $t = 0, 1, \ldots, T - 1$, we maintain a set of (allegedly) good machines $\mathsf{good}_t \subseteq [m]$. We begin with $\mathsf{good}_0 = [m]$ and start to detect malicious machines and remove them from the set. We choose a learning rate $\eta > 0$, and perform the SGD update

$$x_{t+1} = x_t + \xi_t - \eta \frac{1}{|\mathsf{good}_t|}\sum_{i \in \mathsf{good}_t} \nabla_{t,i}$$

where $\xi_t \sim \mathcal{N}(0, \nu^2\mathbf{I})$ is a random Gaussian perturbation that is added for theoretical purpose.

For each machine $i \in [m]$, we keep track of the history of its stochastic gradients up to *two windows*. Namely, $A_i \leftarrow \sum_{k=last_1}^{t} \frac{\nabla_{k,i}}{|\mathsf{good}_k|}$ and $B_i \leftarrow \sum_{k=last_0}^{t} \frac{\nabla_{k,i}}{|\mathsf{good}_k|}$, for windows sizes $T_0 \leq T_1 \leq T$. We compare among remaining machines in $\mathsf{good}_t$, and kick out those ones whose $A_i$ or $B_i$ deviate "more than usual" to construct $\mathsf{good}_{t+1}$. Conceptually, we view these two as *safe guards*.

Our theory makes sure that, when the "window sizes" and the thresholds for "more than usual" are defined properly, then $\mathsf{good}_t$ shall always include good, and the algorithm shall proceed to find approximate local minima. Formally, we have (letting the $\widetilde{O}$ notion to hide polylogarithmic factors)

**Theorem 2.3.** *Let $C_3 = \alpha^2 + \frac{1}{m}$. Suppose we choose $\nu^2 = \widetilde{\Theta}(C_3)$, $\eta = \widetilde{\Theta}(\frac{\varepsilon^2}{dC_3})$, $T_0 = \widetilde{\Theta}(\frac{1}{\eta})$, $T_1 = \widetilde{\Theta}(\frac{1}{\eta\sqrt{\varepsilon}})$, $\mathfrak{T}_0 = \widetilde{\Theta}(\sqrt{T_0})$, and $\mathfrak{T}_1 = \widetilde{\Theta}(\sqrt{T_1})$, then after*

$$T = \widetilde{O}\left(\frac{(f(x_0) - \min f(x))d}{\varepsilon^4}(\alpha^2 + \frac{1}{m})\right)$$

*iterations, with high probability, for at least constant fraction of the indices $t \in [T]$, they satisfy*

$$\|\nabla f(x_t)\| \leq \varepsilon \quad and \quad \nabla^2 f(x_t) \succeq -\sqrt{\varepsilon}\mathbf{I} \ .$$

*Remark* 2.4. If one only wishes to achieve a *significantly simpler goal* — finding first-order critical points $\|\nabla f(x_t)\| \leq \varepsilon$ — the analysis becomes much easier (see Section 3.1). In particular, having one safe guard without perturbation (i.e. $\nu = 0$) suffices, and the iteration complexity reduces to $T = \widetilde{O}\left(\frac{f(x_0) - \min f(x)}{\varepsilon^4}(\alpha^2 + \frac{1}{m})\right)$. Bulusu et al. (2020) achieves this easier goal but requires an additional assumption: there is one guaranteed good worker known by the master.

**Our contribution.** We reiterate our theoretical contributions from three perspectives. 1) When $\alpha < 1/\sqrt{m}$, our algorithm requires $mT = \widetilde{O}\left(\frac{(f(x_0) - \min f(x))d}{\varepsilon^4}\right)$ stochastic gradient computations. This matches the best known result (Jin et al., 2019) under our minimal assumptions of the non-convex objective. (There exist other works in the stochastic setting that break the $\varepsilon^{-4}$ barrier

and get rid of the dimension dependence $d$ under stronger assumptions.)[5]. 2) When $\alpha < 1/\sqrt{m}$, our algorithm enjoys linear parallel speed-up: the parallel time complexity reduces by a factor of $\Theta(m)$. When $\alpha \in [1/\sqrt{m}, 1/2)$, our parallel time complexity is $\widetilde{O}(\alpha^2)$ times that needed when no parallelism is used, still giving noticeable speedup. The $\alpha^2$ factor also appeared in convex Byzantine distributed optimization (and is known to be tight there) (Yin et al., 2018; Alistarh et al., 2018).

**Comparison to (Yin et al., 2019).** Yin et al. (2019) derived three gradient descent-type algorithms to find points with a weaker (and less standard) guarantee: $\|\nabla f(x)\| \leq \varepsilon$ and $\nabla^2 f(x) \succeq -(\varepsilon^2 d)^{1/5}\mathbf{I}$. Despite practical differences (namely, gradient descent may be less favorable comparing to stochastic gradient descent especially in deep learning applications), the parallel time complexities derived from their result are also generally larger than ours.

Their paper focuses on bounding the number of sampled stochastic functions, as opposed to the number of stochastic gradient evaluations like we do. When translated to our language, each of the workers in their setting needs to evaluate $T$ stochastic gradients, where (1) $T = \widetilde{O}\left(\frac{\alpha^2 d}{\varepsilon^4} + \frac{d^2}{\varepsilon^4 m} + \frac{\sqrt{d}}{\varepsilon^3}\right)$ if using coordinate-wise median, (2) $T = \widetilde{O}\left(\frac{\alpha^2 d^2}{\varepsilon^4} + \frac{d^2}{\varepsilon^4 m}\right)$ if using trimmed mean, and (3) $T = \widetilde{O}\left(\frac{\alpha}{\varepsilon^4} + \frac{d}{\varepsilon^4 m}\right)$ if using iterative filtering. The complexities (1) and (2) are larger than ours (also with a weaker guarantee); the complexity (3) seems incomparable to ours, but when translating to the more standard $(\varepsilon, \sqrt{\varepsilon})$ guarantee, becomes $T = \widetilde{O}\left(\frac{\alpha d^2}{\varepsilon^5} + \frac{d^3}{\varepsilon^5 m}\right)$ so is also larger than ours. It is worth noting that (3) requires $\alpha < 1/4$ so cannot withstand half of the machines being Byzantine.

**Resilience against practical attacks.** Our algorithm's filtering is based upon tracking $B_i$ (resp. $A_i$), the stochastic gradients of each machine $i$ *averaged over a window of $T_0$ (resp. $T_1$) iterations.* This is a departure from previous defenses, most of which are history-less, and enables us to be provably Byzantine-resilient against state-of-the-art attacks (Baruch et al., 2019; Xie et al., 2020).

In Baruch et al. (2019), Byzantine workers collude to shift the gradient mean by a factor $\beta$ times the standard deviation of the (true stochastic) gradient, while staying within population variance. They noticed $\beta$ can be quite large especially in neural network training. Their attack circumvent existing defenses because those defense algorithms are "historyless", while their attack is statistically indistinguishable from an honest execution in any single iteration. However, our algorithm can provably defend against this attack since it has *memory*: Byzantine workers following their strategy will progressively diverge from the (honest) "median" $B_{\mathsf{med}}$ (by an amount proportional to $\Omega(T)$ in $T$ iterations as opposed to $\sqrt{T}$), and be marked as malicious by our algorithm. (See Figure 2(a).) In Xie et al. (2020), Byzantine workers deviate in the negative direction of the gradient. However, to avoid being caught by our algorithm, the maximum "magnitude" of this attack has to stay within our thresholds. We implemented both attacks and showed our algorithm's robustness experimentally.

Finally, we note that prior "historyless" schemes, such as Krum or median-based schemes, could be thought of as providing stronger guarantees, as they in theory allow Byzantine nodes to change IDs during the computation: such schemes only require an upper bound on the number of Byzantine agents in each round. However, the attack of Baruch et al. (2019) essentially shows that all such schemes are vulnerable to variance attacks, and that such attacks are eminently plausible in practice. Thus, this suggests that the use of historical information, which requires that Byzantine nodes cannot change their IDs during the execution, may be necessary for Byzantine resilience.

**Tolerating transient failures and node ID relabeling.** Our algorithm can also withstand *transient* node failures and *some degrees of ID relabeling*, by resetting the set of good nodes $\mathsf{good}_t$ to include all nodes every $T_1$ steps. The algorithm then proceeds as usual. The key observation behind this relaxation is the fact that our analysis only requires that the attack conditions hold inside the current window. (Please see the Theorem B.1 for details.) We validate this experimentally in Section 5.

## 3 WARMUP: SINGLE SAFE GUARD

As a warmup, let us first analyze the behavior of perturbed SGD with a single safe guard. Consider Algorithm 2, where we start with a point $w_0$, a set $\mathsf{good}_0 \supseteq \mathsf{good}$, and perform $T$ steps of perturbed SGD. (We use the $w_t$ sequence instead of the $x_t$ sequence to emphasize that we are in Algorithm 2.)

---

[5]Works such as (Allen-Zhu, 2018a; Lei et al., 2017; Tripuraneni et al., 2017; Allen-Zhu, 2018b; Fang et al., 2018; Nguyen et al., 2017) require $f(x) = \mathbb{E}_{s \sim \mathcal{D}}[f_s(x)]$ where each $f_s(x)$ is second-order smooth and/or Lipschitz smooth. This requirement may be too strong for certain practical applications.

---

**Algorithm 2** Perturbed SGD with single safe guard (for analysis purpose only)

---

**Input:** point $w_0 \in \mathbb{R}^d$, set $\mathsf{good}_0 \supseteq \mathsf{good}$, rate $\eta > 0$, length $T \geq 1$, threshold $\mathfrak{T} > 0$;

1: **for** $t \leftarrow 0$ to $T - 1$ **do**
2:     **for each** $i \in \mathsf{good}_t$ **do**
3:         receive $\nabla_{t,i} \in \mathbb{R}^d$ from machine $i$;
4:         $B_i \leftarrow \sum_{k=0}^{t} \frac{\nabla_{k,i}}{|\mathsf{good}_k|}$;
5:     $B_{\mathsf{med}} \leftarrow B_i$ where $i \in \mathsf{good}_t$ is any machine s.t. $\left| \{ j \in \mathsf{good}_t : \|B_j - B_i\| \leq \mathfrak{T} \} \right| > m/2$.
6:     $\mathsf{good}_{t+1} \leftarrow \{ i \in \mathsf{good}_t : \|B_i - B_{\mathsf{med}}\| \leq 2\mathfrak{T} \}$ ;
7:     $w_{t+1} = w_t - \eta \left( \xi_t + \frac{1}{|\mathsf{good}_t|} \sum_{i \in \mathsf{good}_t} \nabla_{t,i} \right)$;       $\diamond$ *Gaussian noise* $\xi_t \sim \mathcal{N}(0, \nu^2 \mathbf{I})$

---

**Definition 3.1.** *We make the following definition to simplify notations: let* $\Xi_t := \sigma_t + \Delta_t$ *where*

- $\sigma_t := \frac{1}{|\mathsf{good}_t|} \sum_{i \in \mathsf{good}} \left( \nabla_{t,i} - \nabla f(w_t) \right)$
- $\Delta_t := \frac{1}{|\mathsf{good}_t|} \sum_{i \in \mathsf{good}_t \setminus \mathsf{good}} \left( \nabla_{t,i} - \nabla f(w_t) \right)$

*Therefore, we can re-write the SGD update as* $w_{t+1} = w_t - \eta(\nabla f(w_t) + \xi_t + \Xi_t)$ .

The following lemma is fairly immediate to prove:

**Lemma 3.2** (single safe guard). *In Algorithm 2, suppose we choose* $\mathfrak{T} = 8\sqrt{T \log(16mT/p)}$. *Then, with probability at least* $1 - p/4$, *for every* $t = 0, \ldots, T - 1$,

- $\mathsf{good}_t \supseteq \mathsf{good}$.
- $\|\sigma_t\|^2 \leq O(\frac{\log(T/p)}{m})$ *and* $\|\sigma_0 + \cdots + \sigma_{t-1}\|^2 \leq O(\frac{T \log(T/p)}{m})$
- $\|\Delta_t\|^2 \leq \alpha^2$ *and* $\|\Delta_0 + \cdots + \Delta_{t-1}\|^2 \leq O(\alpha^2 T \log(mT/p))$
- $|\langle \nabla f(w_t), \xi_t \rangle| \leq \|\nabla f(w_t)\| \cdot O(\nu \sqrt{\log(T/p)})$,
- $\|\xi_t\|^2 \leq O(\nu^2 d \log(T/p))$, $\|\xi_0 + \cdots + \xi_{t-1}\|^2 \leq O(\nu^2 dT \log(T/p))$

*We call this probabilistic event* $\mathsf{Event}_T^{\mathsf{single}}(w_0)$ *and* $\Pr[\mathsf{Event}_T^{\mathsf{single}}(w_0)] \geq 1 - p/4$.

(The third property above is ensured by our choice of $\mathfrak{T}$ and the use of safe guard, and the rest of the properties follow from simple martingale concentration arguments. Details are in Appendix A.1.)

### 3.1 CORE TECHNICAL LEMMA 1: OBJECTIVE DECREASE

Our first main technical lemma is the following:

**Lemma 3.3.** *Suppose we choose* $\mathfrak{T}$ *as in Lemma 3.2. Denote by* $C_1 = \log(T/p)$ *and* $C_2 = \alpha^2 \log \frac{mT}{p} + \frac{\log(T/p)}{m}$. *Suppose* $\eta \leq 0.01 \min\{1, \frac{1}{C_2}\}$, $T = \frac{1}{100\eta(1+\sqrt{C_2})}$ *and we start from* $w_0$ *and apply Algorithm 2. Under event* $\mathsf{Event}_T^{\mathsf{single}}(w_0)$, *it satisfies*

$$f(w_0) - f(w_T) \geq 0.7\eta \sum_{t=0}^{T-1} \left( \|\nabla f(w_t)\|^2 - \eta \cdot O(C_2 + (C_2)^{1.5}) - O(C_1 \nu^2 \eta(d + \sqrt{C_2})) \right)$$

Lemma 3.3 says after $T \approx \frac{1}{\eta}$ steps of perturbed SGD, the objective value decreases by, up to some small additive error and *up to logarithmic factors*, $f(w_0) - f(w_T) \geq 0.7\eta \sum_{t=0}^{T-1}(\|\nabla f(w_t)\|^2 - \eta C_2)$. This immediately implies, if we choose $\eta \approx \frac{\varepsilon^2}{C_2}$, then by repeating this analysis for $O(\frac{C_2}{\varepsilon^4}) = O(\frac{\alpha^2 + 1/m}{\varepsilon^4})$ iterations, we can find approximate critical point $x$ with $\|\nabla f(x)\| \leq \varepsilon$.

*Proof sketch of Lemma 3.3.* The full proof is in Appendix A.2 but we illustrate the main idea and difficulties below. After simple manipulations, it is not hard to derive that

$$f(w_0) - f(w_T) \gtrsim 0.9\eta \sum_{t=0}^{T-1} \left( \|\nabla f(w_t)\|^2 - \eta \right) + \underbrace{\eta \sum_{t=0}^{T-1} \langle \nabla f(w_t), \Xi_t \rangle}_{\text{remainder terms}}$$

where recall that $\Xi_t = \sigma_t + \Delta_t$. When there are no Byzantine machines, we have $\mathbb{E}[\Xi_t] = \mathbb{E}[\sigma_t] = 0$ so the remainder terms must be small by martingale concentration. Therefore, the main technical difficulty arises to deal with those Byzantine machines, who can adversarially design their $\nabla_t$ (even by collusion) so as to negatively correlate with $\nabla f(w_t)$ to "maximally destroy" the above inequality.

Our main idea is to use second-order smoothness to write $\nabla f(w_t) \approx \nabla f(w_0) + \nabla^2 f(w_0) \cdot (w_t - w_0)$. To illustrate our idea, let us ignore the constant vector and assume that the Hessian is the identity: that is, imagine as if $\nabla f(w_t) \approx w_t - w_0$. Using $w_t - w_0 = -\sum_{k<t} \Xi_t + \xi_t$, we immediately have

$$-\langle \nabla f(w_t), \Xi_t \rangle \approx -\langle w_t - w_0, \Xi_t \rangle = \sum_{k<t} \langle \Xi_k, \Xi_t \rangle + \sum_{k<t} \langle \xi_k, \Xi_t \rangle \qquad (3.1)$$

For the first partial sum $\langle \sum_{k<t} \Xi_k, \Xi_t \rangle$ in (3.1), it is easy to bound its magnitude using our safeguard. Indeed, we have $\left| \sum_t \langle \sum_{k<t} \Xi_k, \Xi_t \rangle \right| \leq \| \sum_t \Xi_t \|^2 + \sum_t \| \Xi_t \|^2$ so we can apply Lemma 3.2. For the second partial sum $\sum_t \sum_{k<t} \langle \xi_k, \Xi_t \rangle$, we can apply the concentration Proposition 3.4 below. $\qquad \square$

**Proposition 3.4.** *Fix the dimension parameter $d \geq 1$. Suppose $\xi_0, \ldots, \xi_{T-1} \in \mathbb{R}^d$ are i.i.d. drawn from $\mathcal{N}(0, \mathbf{I})$, and that $\Delta_1, \ldots, \Delta_{T-1}$ are arbitrary vectors in $\mathbb{R}^d$. Here, each vector $\Delta_t$ with $t = 1, \ldots, T-1$ can depend on $\xi_0, \ldots, \xi_{t-1}$ but not on $\xi_t, \ldots, \xi_{T-1}$. Suppose that these vectors satisfy $\| \Delta_1 + \cdots + \Delta_t \|^2 \leq \mathfrak{T}$ for every $t = 1, \ldots, T-1$. Then, with probability at least $1 - p$,*

$$\left| \sum_{t=1}^{T-1} \langle \xi_0 + \cdots + \xi_{t-1}, \Delta_t \rangle \right| \leq O(\sqrt{dT\mathfrak{T} \log(T/p)}) \ .$$

### 3.2 CORE TECHNICAL LEMMA 2: RANDOMNESS COUPLING

Our next technical lemma studies that, if run Algorithm 2 from a point $w_0$ so that the Hessian $\nabla^2 f(w_0)$ has a eigenvalue which is less than $-\delta$ (think of $w_0$ as a saddle point), then with good probability, after sufficiently many iterations, the sequence $w_1, w_2, \ldots, w_T$ shall *escape* from $w_0$ to distance at least $R$ for some parameter $R \approx \delta$. To prove this, motivated by Jin et al. (2017), we study two executions of Algorithm 2 where their randomness are coupled. We then argue that at least one of them has to escape from $w_0$. For any vector $v$, let $[v]_i$ denote the $i$-th coordinate of $v$.

**Lemma 3.5.** *Suppose we choose $\mathfrak{T}$ as in Lemma 3.2 and $C_1, C_2$ as in Lemma 3.3. Suppose $w_0 \in \mathbb{R}^d$ satisfies $\lambda_{\min}(\nabla^2 f(w_0)) = -\delta$ for some $\delta \geq 0$. Without loss of generality let $\mathbf{e}_1$ be the eigenvector of $\nabla^2 f(w_0)$ with smallest eigenvalue. Consider now two executions of Algorithm 2, both starting from $w_0^{\mathsf{a}} = w_0^{\mathsf{b}} = w_0$, and suppose their randomness $\{\xi_t^{\mathsf{a}}\}_t$ and $\{\xi_t^{\mathsf{b}}\}_t$ are coupled so that $[\xi_t^{\mathsf{a}}]_1 = -[\xi_t^{\mathsf{b}}]_1$ but $[\xi_t^{\mathsf{a}}]_i = [\xi_t^{\mathsf{b}}]_i$ for $i > 1$. In words, the randomness is the same orthogonal to $\mathbf{e}_1$, but along $\mathbf{e}_1$, the two have opposite signs. Now, suppose we perform $T = \Theta(\frac{1}{\eta\delta} \log \frac{R^2\delta}{\eta\nu^2})$ steps of perturbed SGD from $w_0^{\mathsf{a}}, w_0^{\mathsf{b}}$ respectively using Algorithm 2. Suppose*

$$R \leq O\big(\frac{\delta}{\sqrt{C_1} \log(R^2\delta/\eta\nu^2)}\big) \quad \text{and} \quad \nu^2 \geq \Omega\big(C_2 \log \frac{R^2\delta}{\eta\nu}\big) \ .$$

*Then, under events $\mathsf{Event}_T^{\mathsf{single}}(w_0^{\mathsf{a}})$ and $\mathsf{Event}_T^{\mathsf{single}}(w_0^{\mathsf{b}})$, with probability at least 0.98, either $\|w_t^{\mathsf{a}} - w_0\| > R$ or $\|w_t^{\mathsf{b}} - w_0\| > R$ for some $t \in [T]$.*

Proof details in Appendix A.4. The main proof difficulty is to analyze a noisy version of the power method, where the noise comes from (1) Gaussian perturbation (which is the good noise), (2) stochastic gradients (which has zero mean), and (3) Byzantine workers (which can be adversarial).

## 4 FROM WARMUP TO FINAL THEOREM WITH DOUBLE SAFE GUARDS

At a high level, Lemma 3.3 ensures that if we keep encountering points with large gradient $\|\nabla f(w_t)\|$, then the objective should sufficiently decrease; in contrast, Lemma 3.5 says that if we keep encountering points with negative Hessian directions (i.e., $\lambda_{\min}(\nabla^2 f(w_t)) < -\delta$), then the points must move a lot (i.e., by more than $R$ in $T$ iterations, which can also lead to sufficient objective decrease, see Lemma B.4). Therefore, at a high level, when the two lemmas are combined, they tell that we *must not* encounter points with $\|\nabla f(x)\|$ being large, or $\lambda_{\min}(\nabla^2 f(x))$ being very negative, for too many iterations. Therefore, the algorithm can find approximate local minima.

The reason we need *two safe guards*, is because the number of rounds $T$ for Lemma 3.3 and Lemma 3.5 differ by a factor. We need two safe guards with different window sizes to ensure the two lemmas simultaneously hold. We encourage the reader to examine the full analysis in Appendix B.

## 5 EXPERIMENTAL VALIDATION

We evaluate the convergence of `SafeguardSGD` to examine its practical performance against prior works. We perform the non-convex task of *training a residual network ResNet-20* (He et al., 2016) on the CIFAR-10/100 datasets (Krizhevsky et al., 2014). More details are given in Appendix C.

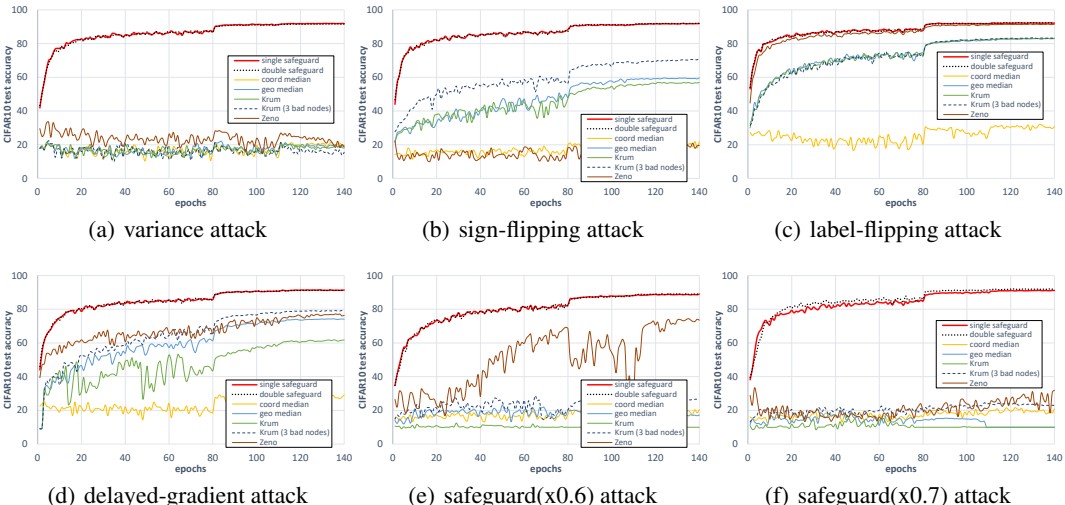

Figure 1: Convergence comparison (CIFAR-10 test accuracy) under different attacks. (In Appendix C.2, one can find additional CIFAR-100 experiments, more discussions, and bigger plots.)

We instantiate $m = 10$ workers and one master node executing data-parallel SGD for 140 passes (i.e. epochs) over the training dataset. The results for higher number of workers and epochs are similar, and therefore omitted. We compare against *Geometric Median* (Chen et al., 2017), *Coordinate-wise Median* (Yin et al., 2018; 2019), *Krum* (Blanchard et al., 2017), and *Zeno* (Xie et al., 2018b). Overall, our experimental setup is very similar to Zeno (Xie et al., 2018b) but with additional attacks.

We implemented the approach of Yang et al. (2019), but found it very sensitive to hyper-parameter values and were unable to make it converge across all attacks even after significant tuning of its $\gamma$ parameter. We also implemented the *convex* algorithm of Alistarh et al. (2018), and executed it in our *non-convex* setting. We found their algorithm can be easily attacked on our ResNet training tasks. There exists a simple attack, described in Appendix C.4 which causes their algorithm to either mislabel most good workers as Byzantine, or diverge, or converge to very poor solutions. This is not surprising, since their algorithm is designed for, and only guaranteed to work in, the convex setting.

To make the comparison stronger, when implementing `SafeguardSGD`, we have chosen fixed window sizes $T_0 = 1$ epoch and $T_1 = 6$ epochs across all experiments, and adopted an automated process to select $\mathfrak{T}_0, \mathfrak{T}_1$. Determining these thresholds requires being able to pre-run the task on an honest worker. We have also implemented a single safeguard variant of `SafeguardSGD`, with window size $T = 3$ epochs.

**Attacks.** We set $\alpha = 0.4$, which means that there are 4 Byzantine workers. (This exceeds the fault-tolerance of Krum, and so we also tested Krum with only 3 Byzantine workers.)

- LABEL-FLIPPING ATTACK: each Byzantine worker computes its gradient based on the cross-entropy loss with flipped labels: for CIFAR-10, label $\ell \in \{0, ..., 9\}$ is flipped to $9 - \ell$.
- DELAYED-GRADIENT ATTACK: each Byzantine worker sends an *old* gradient to master. In our experiments, the delay is of $D = 1000$ iterations.
- VARIANCE ATTACK (Baruch et al., 2019): Byzantine workers measure the mean and the standard-deviation of gradients at each round, and collude to move the mean by the largest value which still operates within population variance. (For our parameter settings, this is $0.3$ times the standard deviation. We discuss results for additional parameter values in the Appendix.)
- SIGN-FLIPPING ATTACK: each Byzantine worker sends the negative gradient to the master.
- SAFEGUARD ATTACK: each Byzantine workers sends a negative but re-scaled gradient to the master. We use re-scale factors $0.6$ and $0.7$ in our experiments. The re-scale factor $0.6$ avoids triggering the safe-guard conditions at the master, and the re-scale factor $0.7$ occasionally triggers the safe-guard conditions. This attack is an instantiation of the inner-product attack (Xie et al., 2020), customized specifically to maximally affect our `SafeguardSGD` algorithm.

**Main experimental results.** The ideal test accuracy is 91.7%, which corresponds to applying SGD using only the stochastic gradients from the honest workers. Figure 1 compares the performances

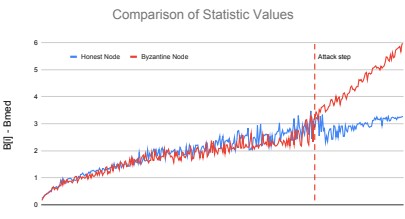

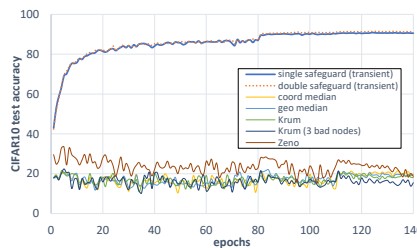

(a) $\|B_i - B_{\text{med}}\|$ between a good node (blue), and a bad node (red) which pretends to be honest and then starts to apply the variance attack.

(b) Convergence for our safeguard algorithms under the variance attack, after periodically resetting the set of good nodes.

Figure 2

in test accuracy. Below we *summarize our main findings* for the experiments, and we defer detailed discussions (and additional experiments for CIFAR-100) to Appendix C.

- `SafeguardSGD` generally outperforms all the previous methods in test accuracy. The test accuracy difference can be "90% vs. < 40%" between our algorithm and the best prior work.
- The *variance attack* is indeed **very strong**, in that it severely affects the accuracy of all prior works (test accuracy < 35%). This is because thesese defenses are "historyless." By contrast, our algorithm not only provably but also *empirically* defends against it.
- Our *safeguard attack* (especially with re-scale factor 0.7) is **as strong as** the variance attack, and even **stronger** on the CIFAR-100 dataset; please see the results in Appendix C.2.5.
- The *label-flipping attack* is **rather weak**: although some defenses, such as Zeno, did not determine which of the workers are malicious, they still converge well under this attack.
- The *sign-flipping* and *delayed-gradient* attacks are **moderate**: the best prior works can achieve accuracy $60\% \sim 70\%$. It is worth noting that the *sign-flipping* attack can already nullify the Zeno defence (test accuracy 20%). The issue seems to be that it can be very hard for Zeno to use relatively few samples to determine if the gradient direction is flipped to negative.
- `SafeguardSGD` can easily **catch** all the bad workers under *sign-flipping* and *variance attacks*, and thus leads to gives ideal performance. It **cannot catch** any bad worker for *label-flipping* and *delayed-gradient attacks*, but there is no performance loss anyways if we use such bad gradients.
- The *safeguard attacks*, designed to *maximally* impact the performance of our `SafeguardSGD`, can indeed affect our performance. Specifically, under re-scale factor 0.6, the test accuracy drops from 91.7% to 89.3% because `SafeguardSGD` **cannot catch** any bad worker; however, under re-scale factor 0.7, the test accuracy no longer drops because `SafeguardSGD` can begin to **catch some** bad workers (it can catch between 0 and 4 bad workers depending on the randomness.)
- In most cases, the single-safeguard algorithm is close to double-safeguard, except for the safeguard(x0.7) attack, in which using double-safeguard one can more easily catch bad workers. (This is more apparent in the CIFAR-100 experiment, see Appendix C.2.5.)

We conclude that `SafeguardSGD` can be practical, and outperforms previous approaches.

**A deeper dive: how the algorithm works.** Let us explain the inner workings of our algorithm in the context of a "delayed" attack, where the Byzantine nodes collude to execute an attack only after a specific, given point in the execution (in this case, the first half-epoch). Figure 2(a) presents the results from the perspective of the value of $\|B_i - B_{\text{med}}\|$ registered at the master server, for two nodes, an honest one, and a Byzantine one. The value of $\|B_i - B_{\text{med}}\|$ increases for all the nodes (at a rate of roughly $\sqrt{t}$ at step $t$); but, once the attack starts, the statistic for the Byzantine node grows *linearly* in $t$, leading to fast detection.

**Transient attacks and node ID relabeling.** Finally, in Figure 2(b) we analyze the behaviour of our algorithm when it periodically (every 3 epochs for single safeguard and 6 epochs for double safeguard) *resets* the set of *good* nodes to include all nodes, restarting the detection process from scratch. Our theoretical result still applies after this relaxation. This relaxation has two benefits. First, it benefits from bad workers that under *transient* failures (e.g., the node fails for 10 epochs but resumes to work correctly after a while), and thus benefits from the data stored on this worker. Second, it can defend against certain degree of node ID relabeling: it supports the case when good and bad workers exchange their IDs every 6 epochs. In Figure 2(b), we see even under the (very strong) variance attack, relaxed safeguard maintains good performance.

ACKNOWLEDGMENTS

F. E. and D. A. were supported by the European Research Council (ERC) under the European Union's Horizon 2020 research and innovation programme (grant agreement No 805223 ScaleML).

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

# APPENDIX

## A   MISSING PROOFS FOR SECTION 3

### A.1   PROOF OF LEMMA 3.2

Recall the following, useful inequality.

**Lemma A.1** (Pinelis' 1994 inequality (Pinelis, 1994))**.** *Let $X_1, \ldots, X_T \in \mathbb{R}^d$ be a random process satisfying $\mathbb{E}[X_t | X_1, \ldots, X_{t-1}] = 0$ and $\|X_t\| \leq M$. Then,* $\Pr\left[\|X_1 + \cdots + X_T\|^2 > 2\log(2/\delta)M^2 T\right] \leq \delta$.

Lemma 3.2 is in fact a direct corollary of the following claim, whose proof is quite classical. Denote by $C = \log(16mT/p)$. Denote by

$$B_i^{(t)} := \frac{\nabla_{0,i}}{|\text{good}_0|} + \cdots + \frac{\nabla_{t-1,i}}{|\text{good}_{t-1}|} \quad \text{and} \quad B_\star^{(t)} := \frac{\nabla f(w_0)}{|\text{good}_0|} + \cdots + \frac{\nabla f(w_{t-1})}{|\text{good}_{t-1}|}.$$

Recall at iteration $t - 1$, Algorithm 2 computes $\{B_1^{(t)}, \ldots, B_m^{(t)}\}$ as well as some $B_{\text{med}}^{(t)} = B_i^{(t)}$ where $i$ is any machine in $\text{good}_{t-1}$ such that at least half of $j \in [m]$ satisfies $\|B_j^{(t)} - B_i^{(t)}\| \leq 8\sqrt{tC}/m$.

**Claim A.2.** *Let $C = \log(16mT/p)$. Then, with probability at least $1 - p/4$, we have*

*(a) for all $i \in \text{good}$ and $t \in [T]$, $\|B_i^{(t)} - B_\star^{(t)}\| \leq 4\sqrt{tC}/m$.*

*(b) for all $t \in [T]$, each $i \in \text{good}$ is a valid choice for $B_{\text{med}}^{(t)} = B_i^{(t)}$.*

*(c) for all $i \in \text{good}$ and $t \in [T]$, $\|B_i^{(t)} - B_{\text{med}}^{(t)}\| \leq 16\sqrt{tC}/m$ and $\|B_\star^{(t)} - B_{\text{med}}^{(t)}\| \leq 12\sqrt{tC}/m$*

*(d) for all $i \in \text{good}$ and $t \in [T]$, we also have $i \in \text{good}_{t+1}$.*

*(e) $\left\| \sum_{i \in \text{good}} \left(B_i^{(t)} - B_\star^{(t)}\right) \right\| \leq O(\sqrt{t\log(T/p)}/\sqrt{m})$.*

*Proof of Claim A.2.* We prove by induction. Suppose the statements hold for $t - 1$ and we now move to $t$.

(a) For each $i \in \text{good}$, note $\mathbb{E}[\nabla_{t,i}] = \nabla_t$ and $\|\nabla_{t,i} - \nabla_t\| \leq 1$. Let $X_t = \frac{\nabla_{t,i} - \nabla_t}{|\text{good}_t|}$, so that $\|X_t\| \leq \frac{1}{|\text{good}_t|} \leq \frac{1}{(1-\alpha)m} \leq \frac{2}{m}$. We can thus apply Lemma A.1 to the $X_t$ and then take a union bound over all $i \in \text{good}$. Thus, with probability at least $1 - \frac{p}{8T}$ we have $\|B_i^{(t)} - B_\star^{(t)}\| \leq 4\sqrt{tC}/m$ for all $i \in \text{good}$. The result follows from a further union bound over $t \in [T]$.

(b) Claim A.2a implies for every $i, j \in \text{good}$ we have $\|B_i^{(t)} - B_j^{(t)}\| \leq 8\sqrt{tC}/m$. Therefore each $i \in \text{good}$ is a valid choice for setting $B_{\text{med}}^{(t)} = B_i^{(t)}$.

(c) This is a consequence of the previous items and the definition of $B_{\text{med}}^{(t)}$.

(d) This is a consequence of the previous item.

(e) We can apply Lemma A.1 with $\{X_1, X_2, \ldots, X_{t|\text{good}|}\} = \{\frac{\nabla_{k,i} - \nabla f(w_k)}{|\text{good}_k|}\}_{k \in [t], i \in \text{good}}$. It holds with probability at least $1 - \frac{p}{8T}$ that $\left\| \sum_{i \in \text{good}} \left(B_i^{(t)} - B_\star^{(t)}\right) \right\| \leq O(\sqrt{t\log(T/p)}/\sqrt{m})$.

$\square$

*Proof of Lemma 3.2.* The property $\text{good}_t \supseteq \text{good}$ is from Claim A.2d.

The property $\|\sigma_t\|^2 \leq O(\frac{\log(T/p)}{m})$ is by standard concentration inequalities for sums of bounded random vectors.

The property $\|\sigma_0 + \cdots + \sigma_{t-1}\|^2 \leq O(\frac{T\log(T/p)}{m})$ is from Claim A.2e.

The property $\|\Delta_t\| \leq \alpha$ is obvious as we have at most $\alpha$ fraction of the bad machines.

The bound on $\|\Delta_0 + \cdots + \Delta_{t-1}\|^2$ can be derived as follows. For every $i \in [m] \setminus \text{good}$, let $t$ be the last iteration $i$ satisfies $i \in \text{good}_t$. Then, by the triangle inequality,

$$\|B_i^{(t+1)} - B_\star^{(t+1)}\| \leq \frac{2}{m} + \|B_i^{(t)} - B_\star^{(t)}\|$$

On the other hand, $t \in \text{good}_t$ implies $\|B_i^{(t)} - B_{\text{med}}^{(t)}\| \leq 16\sqrt{tC}/m$ by the algorithm; combining this with $\|B_\star^{(t)} - B_{\text{med}}^{(t)}\| \leq 12\sqrt{tC}/m$, and summing up over all such bad machines $i$ finishes the proof.

The final two properties follow from standard facts about Gaussian random vectors.

$\square$

## A.2 Proof of Lemma 3.3

*Proof of Lemma 3.3.* Using the Lipschitz smoothness of $f(\cdot)$, we have

$$
\begin{aligned}
f(w_t) - f(w_{t+1}) &\geq \langle \nabla f(w_t), w_t - w_{t+1}\rangle - \frac{1}{2}\|w_t - w_{t+1}\|^2 \\
&= \eta\|\nabla f(w_t)\|^2 + \eta\langle \nabla f(w_t), \Xi_t\rangle - \frac{1}{2}\|w_t - w_{t+1}\|^2 + \eta\langle \nabla f(w_t), \xi_t\rangle
\end{aligned}
$$

We first show:

$$
\|w_t - w_{t+1}\|^2 = \eta^2\|\nabla f(w_t) + \Xi_t - \xi_t\|^2 \leq 3\eta^2(\|\nabla f(w_t)\|^2 + \|\Xi_t\|^2 + \|\xi_t\|^2)
$$

$$
\Big|\sum_{t=0}^{T-1}\eta\langle \nabla f(w_t), \xi_t\rangle\Big| \leq \eta\sqrt{\sum_{t=0}^{T-1}\|\nabla f(w_t)\|^2 \cdot O(\nu\sqrt{C_1})} \leq \left(0.05\eta\sum_{t=0}^{T-1}\|\nabla f(w_t)\|^2\right) + O(C_1\nu^2\eta)
$$

The first follows since $(a+b+c)^2 \leq 3(a^2+b^2+c^2)$ for any $a, b, c \in \mathsf{R}$, and the second follows from Lemma 3.2. Combining them, and also using that $\|\Xi_t\|^2 \leq O(C_2)$, $\|\xi_t\|^2 \leq O(d\nu^2 C_1)$, and $\eta \leq 0.01$, we have

$$
f(w_0) - f(w_T) \geq 0.9\eta\sum_{t=0}^{T-1}\left(\|\nabla f(w_t)\|^2 - O(\eta C_2)\right) + \eta\sum_{t=0}^{T-1}\langle \nabla f(w_t), \Xi_t\rangle - O(\eta T\nu^2 C_1(\eta d + \frac{1}{T}))
\tag{A.1}
$$

For the inner product on the right hand of (A.1), we have that

$$
\eta\sum_{t=0}^{T-1}\langle \nabla f(w_t), \Xi_t\rangle = \underbrace{\frac{\eta}{T}\sum_{q=0}^{T-1}\Big\langle \nabla f(w_q), \sum_{t=0}^{T-1}\Xi_t\Big\rangle}_{\spadesuit} + \underbrace{\frac{\eta}{T}\sum_{q=0}^{T-1}\sum_{t=0}^{T-1}\langle \nabla f(w_t) - \nabla f(w_q), \Xi_t\rangle}_{\clubsuit}
\tag{A.2}
$$

For the first term $\spadesuit$, we have

$$
\begin{aligned}
|\spadesuit| &\leq \frac{\eta}{T}\sum_{q=0}^{T-1}\Big|\Big\langle \nabla f(w_q), \sum_{t=0}^{T-1}\Xi_t\Big\rangle\Big| \leq \frac{\eta}{T}\sum_{q=0}^{T-1}\|\nabla f(w_q)\| \cdot \Big\|\sum_{t=0}^{T-1}\Xi_t\Big\| \\
&\leq 0.1\eta\sum_{q=0}^{T-1}\|\nabla f(w_q)\|^2 + \frac{O(\eta)}{T^2}\sum_{q=0}^{T-1}\Big\|\sum_{t=0}^{T-1}\Xi_t\Big\|^2 \\
&\leq 0.1\eta\sum_{q=0}^{T-1}\|\nabla f(w_q)\|^2 + O\Big(\eta C_2\Big)
\end{aligned}
$$

where the last inequality follows from Lemma 3.2.

For the second term $\clubsuit$, we have

$$
|\clubsuit| \leq \frac{\eta}{T}\sum_{q=0}^{T-1}\Big|\sum_{t=0}^{T-1}\langle \nabla f(w_t) - \nabla f(w_q), \Xi_t\rangle\Big| \leq \underbrace{\frac{\eta}{T}\sum_{q=0}^{T-1}\Big|\sum_{t=0}^{T-1}\langle \nabla^2 f(w_0)(w_t - w_q), \Xi_t\rangle\Big|}_{\diamondsuit}
$$

$$
+ \underbrace{\frac{\eta}{T}\sum_{q=0}^{T-1}\sum_{t=0}^{T-1}(\|w_t - w_0\| + \|w_q - w_0\|)\|w_t - w_q\|\|\Xi_t\|}_{\heartsuit}
$$

Using $\|w_t - w_q\| \leq \|w_t - w_0\| + \|w_q - w_0\|$, one can derive

$$
\begin{aligned}
\heartsuit &\leq \frac{\eta}{T}\sum_{q=0}^{T-1}\sum_{t=0}^{T-1}(\|w_t - w_0\| + \|w_q - w_0\|)^2 \cdot O(\sqrt{C_2}) \\
&\leq \eta\sum_{t=0}^{T-1}\|w_t - w_0\|^2 \cdot O(\sqrt{C_2}) \\
&\leq \eta^3\sum_{t=0}^{T-1}\|\nabla f(w_0) + \cdots + \nabla f(w_{t-1}) + \Xi_0 + \cdots + \Xi_{t-1} + \xi_0 + \cdots + \xi_{t-1}\|^2 \cdot O(\sqrt{C_2}) \\
&\leq O(\sqrt{C_2}\eta^3 T^2)\sum_{t=0}^{T-1}\|\nabla f(w_t)\|^2 + O(\sqrt{C_2}C_2\eta^3 T^2) + O(\eta^3\nu^2 T^2 dC_1\sqrt{C_2})
\end{aligned}
$$

As for $\diamondsuit$,

$$\Big|\sum_{t=0}^{T-1}\langle\nabla^2 f(w_0)(w_t-w_q),\Xi_t\rangle\Big| \leq \Big|\sum_{t=q+1}^{T-1}\langle\nabla^2 f(w_0)(w_t-w_q),\Xi_t\rangle\Big| + \Big|\sum_{t=0}^{q-1}\langle\nabla^2 f(w_0)(w_t-w_q),\Xi_t\rangle\Big|$$

For the first term (and the second term is analogous), we have

$$\Big|\sum_{t=q+1}^{T-1}\langle\nabla^2 f(w_0)(w_t-w_q),\Xi_t\rangle\Big|$$

$$= \eta\Big|\sum_{t=q+1}^{T-1}\langle\nabla^2 f(w_0)(\nabla f(w_q)+\cdots\nabla f(w_{t-1})+\Xi_q+\cdots\Xi_{t-1}+\xi_q+\cdots+\xi_{t-1}),\Xi_t\rangle\Big|$$

$$\leq \eta\Big|\sum_{t=q+1}^{T-1}\langle\nabla^2 f(w_0)(\xi_q+\cdots+\xi_{t-1}),\Xi_t\rangle\Big|+$$

$$\eta\Big|\sum_{t=q+1}^{T-1}\langle\nabla^2 f(w_0)(\nabla f(w_q)+\cdots\nabla f(w_{t-1})),\Xi_t\rangle\Big| + \eta\Big|\sum_{t=q+1}^{T-1}\langle\nabla^2 f(w_0)(\Xi_q+\cdots\Xi_{t-1}),\Xi_t\rangle\Big|$$

$$\overset{\text{①}}{\leq} \eta\cdot O(\sqrt{d\nu^2 TC_1}\cdot\sqrt{TC_2})+$$

$$\eta\Big|\sum_{t=q}^{T-2}\langle\nabla^2 f(w_0)\nabla f(w_t),\Xi_{t+1}+\cdots+\Xi_{T-1}\rangle\Big| + \frac{\eta}{2}\Big\langle\nabla^2 f(w_0)(\Xi_q+\cdots\Xi_{T-1}),(\Xi_q+\cdots\Xi_{T-1})\Big\rangle$$

$$\leq \eta\cdot O(\sqrt{d\nu^2 TC_1}\cdot\sqrt{TC_2}) + \eta\sum_{t=q}^{T-2}\|\nabla f(w_t)\|\|\Xi_{t+1}+\cdots+\Xi_{T-1}\| + \frac{\eta}{2}\|\Xi_q+\cdots\Xi_{T-1}\|^2$$

$$\overset{\text{②}}{\leq} O(\eta\sqrt{TC_2})\cdot\sum_{t=0}^{T-1}\|\nabla f(w_t)\| + O(T\eta C_2+T\eta\nu^2 dC_1)\ .$$

Above, inequality ① uses $\|\Xi_0+\cdots\Xi_t\|\leq O(\sqrt{TC_2})$ for $C_2 = \alpha^2\log\frac{mT}{p} + \frac{\log(T/p)}{m}$ (see Lemma 3.2) and a delicate application of Azuma's inequality that we state at the end of this subsection (see Proposition 3.4); Inequality ② uses Young's inequality and Lemma 3.2.

Putting this back to the formula of $\diamondsuit$, we have

$$\diamondsuit \leq O(\eta^2\sqrt{TC_2})\cdot\sum_{t=0}^{T-1}\|\nabla f(w_t)\| + O(T\eta^2 C_2+T\eta^2\nu^2 dC_1)$$

$$\leq 0.1\eta\sum_{t=0}^{T-1}\|\nabla f(w_t)\|^2 + O(\eta^3 T^2 C_2+T\eta^2 C_2+T\eta^2\nu^2 dC_1)$$

Finally, putting $\diamondsuit$ and $\heartsuit$ back to $\clubsuit$, and putting $\clubsuit$ and $\spadesuit$ back to (A.2) and (A.1), we have

$$f(w_0)-f(w_T) \geq 0.8\eta\sum_{t=0}^{T-1}\|\nabla f(w_t)\|^2 - O(\sqrt{C_2}\eta^3 T^2)\sum_{t=0}^{T-1}\|\nabla f(w_t)\|^2$$

$$- C_2\cdot O(\eta+\eta^2 T+\eta^3 T^2+\sqrt{C_2}\eta^3 T^2) - C_1\cdot O(T\eta^2\nu^2 d+T^2\eta^3\nu^2\sqrt{C_2}+\eta T\nu^2(\eta d+\frac{1}{T}))$$

together with $T = \frac{1}{100\eta(1+\sqrt{C_2})}$ and $\eta\leq 0.01\min\{1,\frac{1}{C_2}\}$, we have

$$f(w_0)-f(w_T) \geq 0.7\eta\sum_{t=0}^{T-1}\|\nabla f(w_t)\|^2 - C_2\cdot O(\eta+\eta^2 T+\eta^3 T^2+\sqrt{C_2}\eta^3 T^2) - C_1\cdot O(T\eta\nu^2\eta(d+\sqrt{C_2}))$$

$$= 0.7\eta\sum_{t=0}^{T-1}\Big(\|\nabla f(w_t)\|^2 - C_2\cdot O(\frac{1}{T}+\eta+\eta^2 T+\sqrt{C_2}\eta^2 T) - C_1\cdot O(\eta T\nu^2\eta(d+\sqrt{C_2}))\Big)$$

$$\geq 0.7\eta\sum_{t=0}^{T-1}\Big(\|\nabla f(w_t)\|^2 - \eta\cdot O(C_2+(C_2)^{1.5}) - O(C_1\nu^2\eta(d+\sqrt{C_2}))\Big)\ .$$

$\square$

## A.3 PROOF OF PROPOSITION 3.4

**Proposition 3.4.** *Fix the dimension parameter $d \geq 1$. Suppose $\xi_0, \ldots, \xi_{T-1} \in \mathbb{R}^d$ are i.i.d. drawn from $\mathcal{N}(0, \mathbf{I})$, and that $\Delta_1, \ldots, \Delta_{T-1}$ are arbitrary vectors in $\mathbb{R}^d$. Here, each vector $\Delta_t$ with $t = 1, \ldots, T-1$ can depend on $\xi_0, \ldots, \xi_{t-1}$ but not on $\xi_t, \ldots, \xi_{T-1}$. Suppose that these vectors satisfy $\|\Delta_1 + \cdots + \Delta_t\|^2 \leq \mathfrak{T}$ for every $t = 1, \ldots, T-1$. Then, with probability at least $1 - p$,*

$$\left| \sum_{t=1}^{T-1} \langle \xi_0 + \cdots + \xi_{t-1}, \Delta_t \rangle \right| \leq O(\sqrt{dT\mathfrak{T} \log(T/p)}) \ .$$

*Proof of Proposition 3.4.* Using the identity formula[6]

$$\sum_{t=1}^{T-1} \langle \xi_0 + \cdots + \xi_{t-1}, \Delta_t \rangle = \left( \sum_{t=0}^{T-2} \xi_t \right) \left( \sum_{t=1}^{T-1} \Delta_t \right) - \sum_{t=1}^{T-2} \langle \xi_t, \Delta_1 + \cdots + \Delta_t \rangle$$

we have

$$\left| \sum_{t=1}^{T-1} \langle \xi_0 + \cdots + \xi_{t-1}, \Delta_t \rangle \right| \leq \left\| \sum_{t=0}^{T-2} \xi_t \right\| \cdot \left\| \sum_{t=1}^{T-1} \Delta_t \right\| + \left| \sum_{t=1}^{T-2} \langle \xi_t, \Delta_1 + \cdots + \Delta_t \rangle \right| \ .$$

$$\leq O(\sqrt{dT\mathfrak{T} \log(T/p)}) + \left| \sum_{t=1}^{T-2} \langle \xi_t, \Delta_1 + \cdots + \Delta_t \rangle \right| \ .$$

where the last inequality uses $\|\xi_0 + \cdots + \xi_{T-2}\| \leq O(\sqrt{dT \log(1/p)})$ with probability at least $1 - p/2$. Furthermore, we note that $\xi_t$ is independent of $\xi_0, \ldots, \xi_{t-1}, \Delta_1, \ldots, \Delta_t$ and $\mathbb{E}[\xi_t] = 0$. Therefore, letting $S_t = \langle \xi_t, \Delta_1 + \cdots + \Delta_t \rangle$, we have $\mathbb{E}[S_t | \xi_0, \ldots, \xi_{t-1}] = 0$; furthermore, with probability at least $1 - p/2$, it satisfies $|S_t| \leq O(\sqrt{d\mathfrak{T} \log(T/p)})$ for every $t$. Finally, by Azuma's inequality, we have

$$\left| \sum_{t=1}^{T-2} \langle \xi_t, \Delta_1 + \cdots + \Delta_t \rangle \right| \leq O(\sqrt{dT\mathfrak{T} \log(T/p)}) \ . \qquad \square$$

## A.4 PROOF OF LEMMA 3.5

*Proof of Lemma 3.5.* Let us denote by $r_t = \frac{[\xi_t^a]_1}{2} = -\frac{[\xi_t^b]_1}{2}$ and we know $r_t \sim \mathcal{N}(0, \frac{\nu^2}{4})$. We can write

$$w_{t+1}^a - w_{t+1}^b = \eta r_t \mathbf{e}_1 + w_t^a - w_t^b - \eta(\nabla f(w_t^a) - \nabla f(w_t^b)) - \eta(\Xi_t^a - \Xi_t^b)$$

Using the second-order smoothness, we have

$$\nabla f(w_t^a) - \nabla f(w_t^b) = \int_{\tau=0}^1 \nabla^2 f\big(w_t^a + \tau(w_t^b - w_t^a)\big)(w_t^a - w_t^b) d\tau$$

$$= \nabla^2 f(w_0) \cdot (w_t^a - w_t^b) + \theta_t$$

for some vector $\|\theta_t\| \leq \max\{\|w_0^a - w_t^a\|, \|w_0^b - w_t^b\|\} \cdot \|w_t^a - w_t^b\|$. Therefore, we have

$$w_{t+1}^a - w_{t+1}^b = \eta r_t \mathbf{e}_1 + \big(\mathbf{I} - \eta \nabla^2 f(w_0)\big)(w_t^a - w_t^b) - \eta(\Xi_t^a - \Xi_t^b + \theta_t)$$

Now, giving $\psi_0 = \overline{\psi}_0 = 0$, imagine two sequences

- $\psi_{t+1} = \eta r_t \mathbf{e}_1 + \big(\mathbf{I} - \eta \nabla^2 f(w_0)\big)\psi_t$ and

- $\overline{\psi}_{t+1} = \eta r_t \mathbf{e}_1 + \big(\mathbf{I} - \eta \nabla^2 f(w_0)\big)\overline{\psi}_t - \eta(\Xi_t^a - \Xi_t^b + \theta_t) = w_{t+1}^a - w_{t+1}^b$

We will inductively prove $\|\psi_t - \overline{\psi}_t\| \leq \frac{1}{2}\|\psi_t\|$. On one hand, it is easy to see that $\psi_t$ is zero except in the first coordinate, in which it behaves as a Gaussian with zero mean and variance $\sum_{k=0}^{t-1}(1 + \eta\delta)^{2k} \cdot \frac{\eta^2 \nu^2}{4} = \Theta\left(\frac{(1+\eta\delta)^{2t}}{\eta\delta} \cdot \eta^2 \nu^2\right)$. By Gaussian tail bounds, we know that

- with probability at least $0.99$, it satisfies $\|\psi_t\| \leq O(\frac{\sqrt{\eta C_1} \nu (1+\eta\delta)^t}{\sqrt{\delta}})$ for every $t$

- with probability at least $0.99$, it satisfies $\|\psi_T\| \geq \frac{1}{1000}(\frac{\sqrt{\eta} \nu (1+\eta\delta)^T}{\sqrt{\delta}})$

In the rest of the proof, we condition on this event happens. We prove towards contradiction by assuming $\|w_t^a - w_0^a\| \leq R$ and $\|w_t^b - w_0^b\| \leq R$ for all $t \in [T]$.

We will inductively prove that $\|\psi_t - \overline{\psi}_t\| \leq \frac{1}{2000}(\frac{\sqrt{\eta} \nu (1+\eta\delta)^t}{\sqrt{\delta}})$. We calculate the difference

$$\psi_t - \overline{\psi}_t = \eta \sum_{i=0}^{t-1} \big(\mathbf{I} - \eta \nabla^2 f(w_0)\big)^{t-1-i}(\Xi_i^a - \Xi_i^b + \theta_i)$$

Let $g = \frac{\psi_t - \overline{\psi}_t}{\|\psi_t - \overline{\psi}_t\|}$, then we can inner-product the above equation by vector $g$, which gives

$$\|\psi_t - \overline{\psi}_t\| = \eta \sum_{i=0}^{t-1} \left\langle \Xi_i^a - \Xi_i^b + \theta_i \, , \, \big(\mathbf{I} - \eta \nabla^2 f(w_0)\big)^{t-1-i} g \right\rangle$$

---

[6]We thank an anonymous reviewer on openreview who pointed out this simpler proof to us.

$$\overset{\text{①}}{\leq} \eta \sum_{i=0}^{t-1} \left( \left\langle \Xi_i^{\mathsf{a}} - \Xi_i^{\mathsf{b}}, \left(\mathbf{I} - \eta \nabla^2 f(w_0)\right)^{t-1-i} g \right\rangle + R \cdot O(\frac{\sqrt{\eta C_1} \nu (1 + \eta \delta)^i}{\sqrt{\delta}}) \cdot (1 + \eta \delta)^{t-1-i} \right)$$

$$\leq \eta \sum_{i=0}^{t-1} \left\langle \Xi_i^{\mathsf{a}} - \Xi_i^{\mathsf{b}}, \left(\mathbf{I} - \eta \nabla^2 f(w_0)\right)^{t-1-i} g \right\rangle + O(R\eta T \frac{\sqrt{\eta C_1}}{\sqrt{\delta}} \nu (1 + \eta \delta)^t)$$

where the inequality ① uses $\|\theta_i\| \leq R \cdot \|\overline{\psi}_i\| \leq R \cdot (\|\psi_i\| + \|\psi_i - \overline{\psi}_i\|)$, $\left\|\left(\mathbf{I} - \eta \nabla^2 f(w_0)\right)^{t-1-i} g\right\| \leq (1 + \eta \delta)^{t-1-i}$, and the inductive assumption. Let us call $M = \left(\mathbf{I} - \eta \nabla^2 f(w_0)\right)$, and focus on

$$\left| \sum_{i=0}^{t-1} \left\langle \Xi_i^{\mathsf{a}}, \left(\mathbf{I} - \eta \nabla^2 f(w_0)\right)^{t-1-i} g \right\rangle \right|$$

$$= \left| \left\langle \Xi_0^{\mathsf{a}} + \cdots + \Xi_{t-1}^{\mathsf{a}}, g \right\rangle + \sum_{i=0}^{t-2} \left\langle \Xi_0^{\mathsf{a}} + \cdots + \Xi_i^{\mathsf{a}}, M^{t-1-i} g - M^{t-2-i} g \right\rangle \right|$$

$$\leq \|\Xi_0^{\mathsf{a}} + \cdots + \Xi_{t-1}^{\mathsf{a}}\| \cdot \|g\| + \sum_{i=0}^{t-2} \|\Xi_0^{\mathsf{a}} + \cdots + \Xi_i^{\mathsf{a}}\| \cdot \|M^{t-1-i} g - M^{t-2-i} g\|$$

$$\leq O(\sqrt{TC_2}) \left( \|g\| + \sum_{i=0}^{t-2} \|M^{t-1-i} g - M^{t-2-i} g\| \right) \qquad \text{(using Lemma 3.2)}$$

$$\leq O(\sqrt{TC_2} \cdot (1 + \eta \delta)^{t-1})$$

Together, we have

$$\|\psi_t - \overline{\psi}_t\| \leq O(\eta \sqrt{TC_2}) \cdot (1 + \eta \delta)^t + O(R\eta T \frac{\sqrt{\eta C_1}}{\sqrt{\delta}} \nu (1 + \eta \delta)^t)$$

Under our assumption, we have $\|\psi_t - \overline{\psi}_t\| < \frac{1}{2000} (\frac{\sqrt{\eta} \nu (1 + \eta \delta)^t}{\sqrt{\delta}})$ and therefore $\|\overline{\psi}_T\| \geq \|\psi_T\| - \|\psi_T - \overline{\psi}_T\| \geq \frac{1}{2000} (\frac{\sqrt{\eta} \nu (1 + \eta \delta)^t}{\sqrt{\delta}})$. Thus, within $T$ iterations, we have $\|\overline{\psi}_t\| > R$ and this gives a contradiction. $\qquad \square$

## B  FINAL: DOUBLE SAFE GUARD

We now come to our final Algorithm 3 which is our perturbed SGD algorithm with two safeguards. The two safeguard algorithm naturally divides itself into epochs, each consisting of $T_1$ iterations. We will demonstrate that within most epochs, we make good progress. Thus, consider some iterate $x_{mT_1}$, for some $m < T/T_1$. Our goal will be to argue that we make good function value progress by iterate $x_{(m+1)T_1}$, and that we do not settle into any saddle points. To slightly simplify notation, let $w_0 = x_{mT_1}$, and let the sequence of iterates be $w_0, \ldots, w_{T_1-1}$, so that $w_{T_1-1} = x_{(m+1)T_1-1}$. For completeness' sake we rewrite this as Algorithm 1.

---

**Algorithm 3** Perturbed SGD with double safe guard (for analysis purpose)

---

**Input:** $w_0 \in \mathbb{R}^d$, set $\mathsf{good}_0 \supseteq \mathsf{good}$, rate $\eta > 0$, lengths $T_1 \geq T_0 \geq 1$, threshold $\mathfrak{T}_1 > \mathfrak{T}_0 > 0$;

1: **for** $t \leftarrow 0$ to $T_1 - 1$ **do**
2:     $last \leftarrow \max\{t_0 \in [t] : t_0 \text{ is a multiple of } T_0\}$
3:     **for each** $i \in \mathsf{good}_t$ **do**
4:         receive $\nabla_{t,i} \in \mathbb{R}^d$ from machine $i$;
5:         $A_i \leftarrow \sum_{k=0}^{t} \frac{\nabla_{k,i}}{|\mathsf{good}_k|}$ and $B_i \leftarrow \sum_{k=last}^{t} \frac{\nabla_{k,i}}{|\mathsf{good}_k|}$;
6:     $A_{\mathsf{med}} \leftarrow A_i$ where $i \in \mathsf{good}_t$ is any machine s.t. $\left|\{j \in \mathsf{good}_t : \|A_j - A_i\| \leq \mathfrak{T}_1\}\right| > m/2$.
7:     $B_{\mathsf{med}} \leftarrow B_i$ where $i \in \mathsf{good}_t$ is any machine s.t. $\left|\{j \in \mathsf{good}_t : \|B_j - B_i\| \leq \mathfrak{T}_0\}\right| > m/2$.
8:     $\mathsf{good}_{t+1} \leftarrow \left\{i \in \mathsf{good}_t : \|A_i - A_{\mathsf{med}}\| \leq 2\mathfrak{T}_1 \bigwedge \|B_i - B_{\mathsf{med}}\| \leq 2\mathfrak{T}_0\right\}$;
9:     $w_{t+1} = w_t - \eta \left(\xi_t + \frac{1}{|\mathsf{good}_t|} \sum_{i \in \mathsf{good}_t} \nabla_{t,i}\right)$;

---

Our main result is the following theorem.

**Theorem B.1.** *Let* $C_3 = \alpha^2 + \frac{1}{m}$. *Suppose we pick parameters* $p, \delta \in (0, 1)$, $\eta \leq \widetilde{O}(\frac{\delta^3}{C_3})$, $\nu^2 = \widetilde{\Theta}(C_3)$, $T_0 = \widetilde{\Theta}(\frac{1}{\eta})$, $T_1 = \widetilde{\Theta}(\frac{1}{\eta\delta}) \geq T_0$, $\mathfrak{T}_1 = \widetilde{\Theta}(\sqrt{T_1})$, *and* $\mathfrak{T}_0 = \widetilde{\Theta}(\sqrt{T_0})$. *Then, starting from* $w_0$,

(a) with probability at least $1 - p$ we have

$$f(w_0) - f(w_{T_1}) \geq 0.7\eta \sum_{t=0}^{T_1-1} \left( \|\nabla f(w_t)\|^2 - \widetilde{O}(\eta C_3 d) \right) \ .$$

(b) As long as $\|w_t - w_0\| \geq R$ for some $t \in \{1, 2, \ldots, T_1\}$ and for $R = \widetilde{\Theta}(\delta) \leq \frac{\delta}{2}$, then with probability at least $1 - p$ we have then

$$f(w_0) - f(w_{T_1}) \geq 0.5\eta \sum_{t=0}^{T_1-1} \left( -\widetilde{O}(\eta C_3 d) \right) + \widetilde{\Omega}(\delta^3)$$

(c) if $\lambda_{\min}(\nabla^2 f(w_0)) \leq -\delta$, we also have with probability at least $0.45$,

$$f(w_0) - f(w_{T_1}) \geq 0.5\eta \sum_{t=0}^{T_1-1} \left( -\widetilde{O}(\eta C_3 d) \right) + \widetilde{\Omega}(\delta^3)$$

## B.1 WHY THEOREM B.1 IMPLIES THEOREM 2.3

Using the parameter choice $\eta = \widetilde{\Theta}(\frac{\varepsilon^2}{C_3 d})$ from Theorem 2.3, we know $\widetilde{O}(\eta C_3 d) \leq 0.1\varepsilon^2$. We claim two things:

- For at least 90% of the epochs, they must satisfy (denoting by $w_0$ and $w_{T_1}$ the beginning and ending points of this epoch)

$$f(w_0) - f(w_{T_1}) \leq 20 \frac{f(x_0) - \min f(x)}{T/T_1} \leq \varepsilon^{1.5}$$

  The last inequality uses our choice of $T$ and $\delta = \widetilde{\Theta}(\sqrt{\varepsilon})$.
  The reason for this is by way of contradiction. Suppose for at least 10% of the epochs it satisfies $f(w_0) - f(w_{T_1}) > 20\frac{f(x_0)-\min f(x)}{T/T_1}$, then, for the remainder of the epochs, they must at least satisfy $f(w_0) - f(w_{T_1}) \geq -0.7\eta T_1 \cdot 0.1\varepsilon^2$. Summing over all the epochs, we shall obtain $f(x_0) - f(x_T) > f(x_0) - \min f(x)$ but this gives a contradiction.
- For at least 40% of the epochs, they must satisfy the three properties from Theorem B.1.

In particular, for at least 30% of the epochs, they must satisfy both. Since $\varepsilon^{1.5}$ is so small that

$$\varepsilon^{1.5} \geq f(w_0) - f(w_{T_1}) \geq 0.5\eta \sum_{t=0}^{T_1-1} \left( -\widetilde{O}(\eta C_3 d) \right) + \widetilde{\Omega}(\delta^3) \geq \widetilde{\Omega}(\delta^3) - 0.05\eta T_1 \varepsilon^2$$

would give a contradiction (for instance, one can choose $\delta$ to be slightly larger than $\sqrt{\varepsilon}$ by some log factors), this means, for those 30% of the epochs, they must satisfy:

- $\varepsilon^{1.5} \geq 0.7\eta \sum_{t=0}^{T_1-1} \left( \|\nabla f(w_t)\|^2 - 0.1\varepsilon^2 \right)$ ,
- $\|w_t - w_0\| \leq \frac{\delta}{2}$ for every $t = 1, 2, \ldots, T_1$, and
- $\nabla^2 f(w_0) \succeq -\delta \mathbf{I}$.

The latter two properties together implies $\nabla^2 f(w_t) \succeq -\frac{\delta}{2}\mathbf{I}$ for every $t = 1, 2, \ldots, T_1$ (by the second-order smoothness). The first property implies for at least 90% of the iterations $t$ in this epoch, they must satisfy $\|\nabla f(x)\| \leq \varepsilon$. This finishes the proof of Theorem 2.3. ∎

## B.2 PROOF OF THEOREM B.1

We first have the following lemma

**Lemma B.2** (double safe guard). *In Algorithm 3, suppose $\mathfrak{T}_1 = 8\sqrt{T_1 \log(16mT_1/p)}$ and $\mathfrak{T}_0 = 8\sqrt{T_0 \log(16mT_1/p)}$. Then, with probability at least $1 - p/2$, for every $t = 0, \ldots, T_1 - 1$,*

- $\text{good}_t \supseteq \text{good}.$
- $\|\sigma_t\|^2 \leq O(\frac{\log(T_1/p)}{m}), \|\Delta_t\|^2 \leq \alpha^2, \|\xi_t\|^2 \leq O(\nu^2 d \log(T_1/p)),$
- $\|\sigma_0 + \cdots + \sigma_{t-1}\|^2 \leq O(\frac{T_1 \log(T_1/p)}{m}), \|\sigma_{last} + \cdots + \sigma_{t-1}\|^2 \leq O(\frac{T_0 \log(T_1/p)}{m})$
- $\|\Delta_0 + \cdots + \Delta_{t-1}\|^2 \leq O(\alpha^2 T_1 \log(mT_1/p))$ and $\|\Delta_{last} + \cdots + \Delta_{t-1}\|^2 \leq O(\alpha^2 T_0 \log(mT_1/p))$
- $\|\xi_0 + \cdots + \xi_{t-1}\|^2 \leq O(\nu^2 d T_1 \log(T_1/p))$ and $\|\xi_{last} + \cdots + \xi_{t-1}\|^2 \leq O(\nu^2 d T_0 \log(T_1/p)).$

*We call this probabilistic event $\text{Event}_{T_1,T_0}^{\text{double}}(w_0)$ and $\Pr[\text{Event}_{T_1,T_0}^{\text{double}}(w_0)] \geq 1 - p/2.$*

The proof is a direct corollary of Lemma 3.2, by combining events $\text{Event}_{T_1}^{\text{single}}(w_0)$, $\text{Event}_{T_0}^{\text{single}}(w_0)$,

$\mathsf{Event}^{\mathrm{single}}_{T_0}(w_{T_0})$, $\mathsf{Event}^{\mathrm{single}}_{T_0}(w_{2T_0})$ and so on. The next lemma is a simple corollary by repeatedly applying Lemma 3.3. It proves Theorem B.1a.

**Lemma B.3** (modified from Lemma 3.3). *Denote by $C_1 = \log(T_1/p)$ and $C_2 = \alpha^2 \log \frac{mT_1}{p} + \frac{\log(T_1/p)}{m}$. Suppose $\eta \leq 0.01 \min\{1, \frac{1}{C_2}\}$, $T_0 = \frac{1}{100\eta(1+\sqrt{C_2})}$ and $T_1 \geq T_0$. We start from $w_0$ and apply Algorithm 3. Under event $\mathsf{Event}^{\mathrm{double}}_{T_1,T_0}(w_0)$, it satisfies*

$$f(w_0) - f(w_{T_1}) \geq 0.7\eta \sum_{t=0}^{T_1-1} \left( \|\nabla f(w_t)\|^2 - \eta \cdot O(C_2 + (C_2)^{1.5}) - O(C_1\nu^2\eta(d + \sqrt{C_2})) \right)$$

The next lemma can be easily derived from Lemma 3.5.

**Lemma B.4** (modified from Lemma 3.5). *Suppose*

$$R = \Theta(\frac{\delta}{\sqrt{C_1}\log(\delta^3/\eta C_2)}) \quad and \quad \nu^2 = \Theta(C_2 \log \frac{\delta^3}{\eta C_2})$$

*Suppose $\eta \leq 0.01 \min\{1, \frac{\delta^3}{C_2}\}$, $T_0 = \frac{1}{100\eta(1+\sqrt{C_2})}$ and $T_1 = \Theta(\frac{1}{\eta\delta} \log \frac{\delta^3}{\eta C_2}) \geq T_0$. Let $w_0 \in \mathbb{R}^d$ be any point in the space and suppose $\lambda_{\min}(\nabla^2 f(w_0)) \leq -\delta$ for some $\delta \geq 0$. Given two coupled sequences defined as before, under events $\mathsf{Event}^{\mathrm{double}}_{T_1,T_0}(w_0^{\mathsf{a}})$ and $\mathsf{Event}^{\mathrm{double}}_{T_1,T_0}(w_0^{\mathsf{b}})$, we have with probability at least 0.98*

$$\max\left\{ f(w_0^{\mathsf{a}}) - f(w_{T_1}^{\mathsf{a}}), f(w_0^{\mathsf{b}}) - f(w_{T_1}^{\mathsf{b}}) \right\}$$

$$\geq 0.5\eta \sum_{t=0}^{T_1-1} \left( -\eta \cdot O(C_2 + (C_2)^{1.5}) - O(C_1\nu^2\eta(d+\sqrt{C_2})) \right) + \Omega(\frac{\delta^3}{C_1 \log^3 \frac{\delta^3}{\eta C_2}})$$

Lemma B.4 directly proves the second half of Theorem B.1c, because given two coupled sequences with the same marginal distribution, we have

$$\Pr[f(w_0^{\mathsf{a}}) - f(w_{T_1}^{\mathsf{a}}) \geq X] \geq \frac{1}{2}\Pr[\max\left\{ f(w_0^{\mathsf{a}}) - f(w_{T_1}^{\mathsf{a}}), f(w_0^{\mathsf{b}}) - f(w_{T_1}^{\mathsf{b}}) \right\} \geq X]$$

*Proof of Lemma B.4.* Our choice on $r$ and $R$ satisfy the requirement of Lemma 3.5. Suppose without loss of generality that the $w_t^{\mathsf{a}}$ sequence leaves $w_0$ by more than $R$. Let $T_1^{\mathsf{a}}$ be the first iteration $t \leq T_1$ in which $\|w_t^{\mathsf{a}} - w_0^{\mathsf{a}}\| \geq R$.

$$\|w_{T_1^{\mathsf{a}}}^{\mathsf{a}} - w_0^{\mathsf{a}}\|^2 = \eta^2 \|\nabla f(w_0^{\mathsf{a}}) + \cdots + \nabla f(w_{T_1^{\mathsf{a}}-1}^{\mathsf{a}}) + \Xi_0^{\mathsf{a}} + \cdots + \Xi_{T_1^{\mathsf{a}}-1}^{\mathsf{a}} + \xi_0^{\mathsf{a}} + \cdots + \xi_{T_1^{\mathsf{a}}-1}^{\mathsf{a}}\|^2$$

$$\leq O(\eta^2 T_1) \sum_{t=0}^{T_1^{\mathsf{a}}-1} \|\nabla f(w_t^{\mathsf{a}})\|^2 + O(C_2\eta^2 T_1) + O(C_1\eta^2\nu^2 T_1 d)$$

Combining this with Lemma B.3, we have

$$f(w_0^{\mathsf{a}}) - f(w_{T_1^{\mathsf{a}}}^{\mathsf{a}}) \geq 0.5\eta \sum_{t=0}^{T_1^{\mathsf{a}}-1} \left( \|\nabla f(w_t^{\mathsf{a}})\|^2 - \eta \cdot O(C_2 + (C_2)^{1.5}) - O(C_1\nu^2\eta(d+\sqrt{C_2})) \right) + \frac{\|w_{T_1^{\mathsf{a}}}^{\mathsf{a}} - w_0^{\mathsf{a}}\|^2}{100\eta T_1}$$

$$\geq 0.5\eta \sum_{t=0}^{T_1^{\mathsf{a}}-1} \left( \|\nabla f(w_t^{\mathsf{a}})\|^2 - \eta \cdot O(C_2 + (C_2)^{1.5}) - O(C_1\nu^2\eta(d+\sqrt{C_2})) \right) + \frac{R^2}{100\eta T_1}$$

Combining this with Lemma B.3 again but for the remainder iterations, we have

$$f(w_0^{\mathsf{a}}) - f(w_{T_1}^{\mathsf{a}}) \geq 0.5\eta \sum_{t=0}^{T_1-1} \left( \|\nabla f(w_t^{\mathsf{a}})\|^2 - \eta \cdot O(C_2 + (C_2)^{1.5}) - O(C_1\nu^2\eta(d+\sqrt{C_2})) \right) + \frac{R^2}{100\eta T_1}$$

$\square$

In fact, the above same proof of Lemma B.4 also implies Theorem B.1b. These together finish the proof of Theorem B.1. ∎

## C   MORE ON EXPERIMENTS

We conduct experiments on training a residual network ResNet-20 He et al. (2016) on the CIFAR-10/100 image classification tasks Krizhevsky et al. (2014).

## C.1   SETTING AND IMPLEMENTED METHODS

In all of our experiments, we use 10 workers and mini-batch size 10 per worker. Given any attacker and any defender algorithm, we run SGD three times for 140 epochs, each time with a different initial learning rate $\eta \in \{0.1, 0.2, 0.4\}$.[7] We let the learning rate decrease by a factor of 10 on epochs 80 and 110, and present present the best testing accuracies in the three runs (each corresponding to a different initial learning rate).

We use standard data augmentation (random crops, random flips, and channel normalization).

We compare against *Geometric Median* Chen et al. (2017), *Coordinate-wise Median* Yin et al. (2018; 2019), *Krum* Blanchard et al. (2017), and *Zeno* Xie et al. (2018b) with attacks. We set $\alpha = 0.4$ so there are 4 Byzantine workers. (This exceeds the fault-tolerance of Krum, and so we also tested Krum with only 3 Byzantine workers.) We formally define those prior works as follows.

**Definition C.1** (GeoMed Chen et al. (2017))**.**   *The geometric median of* $\{y_1, ..., y_m\}$*, denoted by* $geo\_med\{y_1, ..., y_m\}$*, is*

$$geo\_med\{y_1, ..., y_m\} := \arg\min_{y \in \mathbb{R}^d} \sum_{i=1}^{m} \|y - y_i\|$$

*In our experiments, we choose the geometric median from set* $\{y_1, ..., y_m\}$*.*

**Definition C.2** (coordinate-wise median Yin et al. (2018; 2019))**.**   *Coordinate-wise median* $g = med\{y_1, ..., y_m\}$ *is defined as a vector with its $k$-th coordinate being* $g[k] = \boldsymbol{med}\{y_1[k], ..., y_m[k]\}$ *for each* $k \in [d]$*, where* $\boldsymbol{med}$ *is the usual (one-dimensional) median.*

**Definition C.3** (Krum Blanchard et al. (2017))**.**

$$KR\{y_1, ..., y_m\} := y_k \quad where \quad k = \arg\min_{i \in [m]} \sum_{i \to j} \|y_i - y_j\|^2$$

*and* $i \to j$ *is the indices of the* $m - b - 2$ *nearest neighbours of* $y_i$ *in* $\{y_1, ..., y_m\} \setminus \{y_i\}$ *by Euclidean distances.*

Note that Krum requires $2b + 2 < m$. So, we have also repeated the experiments for Krum with 3 Byzantine workers (out of 10 workers) to be more fair.

**Definition C.4** (Zeno Xie et al. (2018b))**.**

$$Zeno_b\{y_1, ..., y_m\} = \frac{1}{m-b} \sum_{i=1}^{m-b} \widetilde{y}(i)$$

*where* $\{\widetilde{y}(i) : i \in [m]\}$ *are the gradient estimators with the* $m - b$ *highest "scores", and the so-called stochastic descendant score for any gradient estimator* $u$*, based on the current parameter $x$, learning rate $\eta$, and a constant weight $\rho > 0$, is defined as:*

$$Score_{\eta, \rho}(u, x) = f_r(x) - f_r(x - \eta u) - \rho \|u\|^2$$

$f_r(x) - f_r(x - \eta u)$ *is the estimated descendant of the loss function and* $\rho \|u\|^2$ *is the magnitude of the update.*

In our experiments, we let $f_r(x)$ be the estimated objective over a mini-batch of size $n_r = 10$ (so the time to perform this estimation is on the same magnitude as the gradient evaluation for each individual worker). We also chose $\rho = 0.0005$ (and this value does not affect our experimental results by much).

**Safeguard SGD.**   Our Algorithm 1 is stated in a way to make our theoretical proofs as clean as possible. Here, we discuss how we actually implement it in practice.

First of all, as common in the literature, we omit the Gaussian noise $\xi_t$ that is added for theoretical purpose, and instead rely on the natural noise in the training process to escape saddle points.

Also, we make universal choices for our safeguard window sizes (across *all* attackers): for our algorithm with a single safeguard we have used a universal window size $T = 3$ epochs, and for our algorithm with double safeguards we have used window sizes $T_0 = 1$ epoch and $T_1 = 6$ epochs.

We also provide an automatic empirical process to select safeguard thresholds and eliminate bad workers.[8] The process to determine $A_{\mathsf{med}}$ (and likewise for $B_{\mathsf{med}}$) is described as follows. In each iteration, for every worker $i \in [m]$, we sort $\left\{\|A_i - A_j\|\right\}_{j \in [m]}$ and pick the smallest $\lceil m/2 + 1 \rceil$-th entry, and let this number be the "score" for worker $i$. We select the worker with the smallest "score" as $A_{\mathsf{med}}$ and call its "score" $S$. Then, we use $1.5 \min\{S, 5\}$ as the safeguard threshold for this iteration. Namely, we declare any worker $j$ satisfying

---

[7]Recall a typical suggested initial learning rate is 0.1 for training ResNet with SGD+momentum; since we are using SGD without momentum, the initial learning rate can be appropriately enlarged.

[8]In our first version of the paper, we pre-run the algorithm for 20 epochs to determine safeguard thresholds; in the newer version, we have avoided the pre-run.

$\|A_j - A_{\text{med}}\| \geq 1.5 \max\{S, 5\}$ as a bad worker.[9]

## C.2 EXPERIMENT RESULTS BY ATTACKS

The *ideal accuracies* are 91.7% / 68.0% for CIFAR-10/100, which correspond to applying SGD using only the stochastic gradients from the honest workers. Below we discuss about the experimental results one attack at a time.

### C.2.1 VARIANCE ATTACK

We begin by looking at the *hardest proposed attack* from prior works. The *Variance attack* follows the strategy prescribed by Baruch et al. (2019), by which Byzantine workers collude in order to shift the mean among all gradients by a factor $\beta$ times the standard deviation of the gradient, while staying within population variance. More precisely, the maximal change to the mean that can be applied by an attacker without the fear of being detected, by using the properties of the Normal distribution, specifically the cumulative standard normal function, and compute the maximal possible shift so that the attackers's values stay within population variance. (See (Baruch et al., 2019, Algorithm 3) for a precise description. Our $\beta$ is $z_{\max}$ in their notation.) We implement this strategy coordinate-wise, the same way as they did. Their work observes that the shift $\beta$ can be non-trivial in practice, since stochastic gradients tend to have large variance in neural network training (which we also observed in our setup). Critically, the attack cannot be defended against by *historyless algorithms*, as the attacker's values are statistically indistinguishable from a regular execution in a single iteration.

In our setting, for 10 total nodes and $\alpha = 0.4$, $\beta$ is upper bounded by 0.3 (following the classic tables for the cumulative normal). We also ran the same attack in the setup from their paper (50 nodes total, of which 24 are Byzantine, which allows $\beta \sim 1.5$) and observed a similar outcome. Results for this experiment are given in Figure 3.

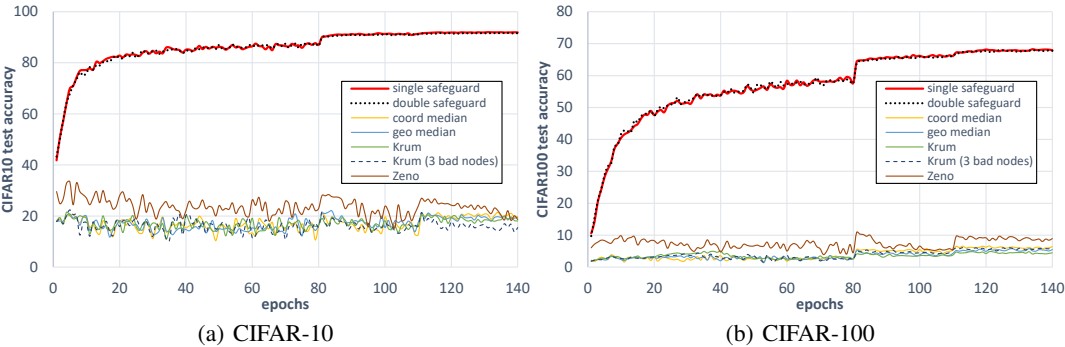

(a) CIFAR-10        (b) CIFAR-100

Figure 3: Performance comparison under the variance attack.

As shown by the results, our algorithm provably circumvents this attack, and recovers full accuracy. This is explained by the fact that the algorithm has *memory*: in particular, Byzantine nodes following this strategy will progressively diverge from the (honest) "median" $A_{\text{med}}$ (at a "linear" rate, recall Figure 2(a)), and therefore will eventually exceed the threshold and be marked as malicious by the algorithm.

Specifically, both variants of the algorithm successfully *catch all the bad nodes* after at most 150 iterations. Indeed, at the 100-th iteration, the pair-wise distances $\|A_i - A_j\|$ among good workers $i, j \in \text{good}$ are between 5.3 and 6.3, but the pair-wise distance between a good and a bad worker is at least 12.5.

### C.2.2 SIGN-FLIPPING ATTACK

We next move onto the *sign-flipping* attack. Recall that, in a sign-flipping attack, each Byzantine worker sends the negative gradient to the master. This is still a strong attack since if one does not avoid any bad workers, the test accuracy will suffer from a significant drop. The results are in Figure 4.

From the plots, one can see that again our single and double safe-guard algorithms *both* outperform prior works. They also successfully *catch all the bad workers* within 150 iterations. (For instance, at iteration 150 for CIFAR-10 training, the distance $\|A_{\text{med}} - A_j\|$ for a good worker $j \in \text{good}$ is at most 6.9, but for a bad

---

[9]This process appears a little different from Algorithm 1, but a similar proof also holds for this new empirical process. The constant factor 1.5 requires no tuning, and the constant threshold 5 is chosen so that the stochastic gradient of batch size 500 at random initialization has Euclidean norm no more than 5.

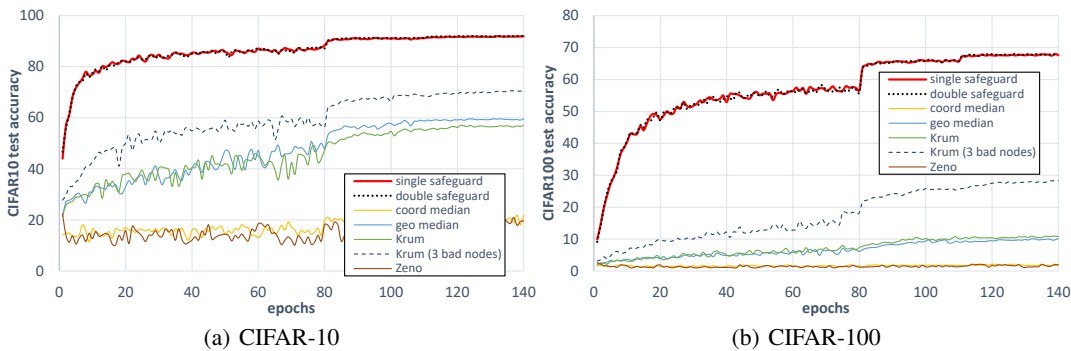

Figure 4: Performance comparison under the sign-flipping attack.

worker $j \notin$ good it can be more than 11.)

In contrast, prior work Zeno completely fails because locally at a training step, using merely $n_r = 10$ samples to evaluate the objective, it is statistically not possible to even distinguish if the sign of the stochastic gradient is flipped or not. For prior works Krum and GeoMedian, although they *appear to* have some non-negligible performances, but they are actually *no better than* simply applying SGD with the naive mean of gradients from all the workers (including those from bad workers).[10] Therefore, we conclude that prior works all fail to be Byzantine fault tolerant under this attack.

### C.2.3 DELAYED-GRADIENT ATTACK

Recall that, in a delayed-gradient attack, each Byzantine worker sends an *old* gradient to the master. In our experiments, the delay is of $D = 1000$ iterations (= 2 epochs). We believe this is not a very strong attack, because delayed gradients are not sufficiently malicious: they are still "correct" to certain extent albeit being delayed. The results are shown in Figure 5.

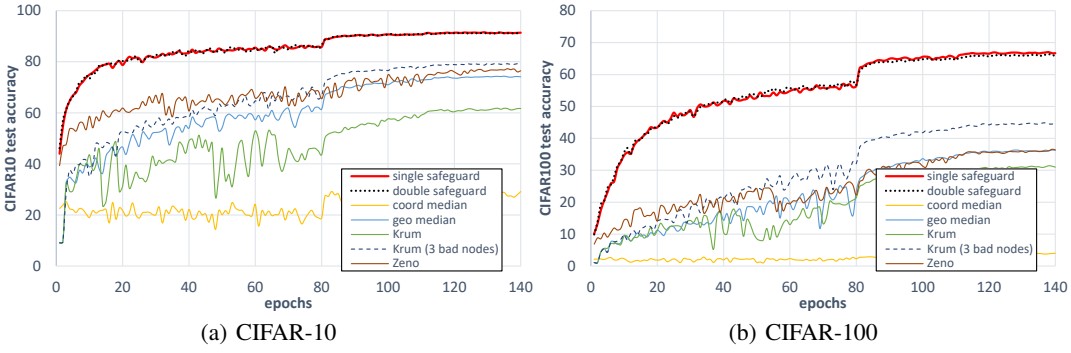

Figure 5: Performance comparison under the delayed-gradient attack.

From the plots, one can see that our single and double safe-guard algorithms again *both* match the ideal accuracies. All the prior works suffer from a significant performance loss under this attack.

It is worth noting that our single and double safe-guard algorithms *do not catch* any bad worker under this attack, so they simply use the "naive mean" of gradients from all the workers (including those delayed gradients from bad workers). However, there is no performance loss even if we use those delayed gradients. That is why we believe the delayed-gradient attack is not very strong, as the gradients are not sufficiently malicious.

Prior work Zeno suffers from some performance loss, because it only uses 6 workers out of 10, in which statistically only $6 \times 0.6 \approx 3 \sim 4$ gradients are correct.[11] Other prior works suffer from performance loss,

---

[10]We did not include this "naive mean" algorithm in the plots for cleanness, but under the sign-flipping attack, it gives 81.4% test accuracy on CIFAR-10 and 38.3% on CIFAR-100. (This should not be surprising, since using $(1 - \alpha)m = 6$ positive gradients plus $\alpha m = 4$ negative gradients still gives non-negligible information about the true gradient.)

[11]In fact, we observed Zeno slightly favors delayed gradients, where each delay gradient is chosen with probability 63%, comparing to true stochastic gradients each chosen with probability 58%.

because they only pick one single stochastic gradient from the 10 workers, and it is sometimes even from the bad worker.

### C.2.4    LABEL-FLIPPING ATTACK

Recall that, in the label-flipping attack, each Byzantine worker computes its gradient based on the cross-entropy loss with flipped labels: for CIFAR-10, label $\ell \in \{0, ..., 9\}$ is flipped to $9 - \ell$, and for CIFAR-100, label $\ell$ is flipped to $99 - \ell$. The results are shown in Figure 6.

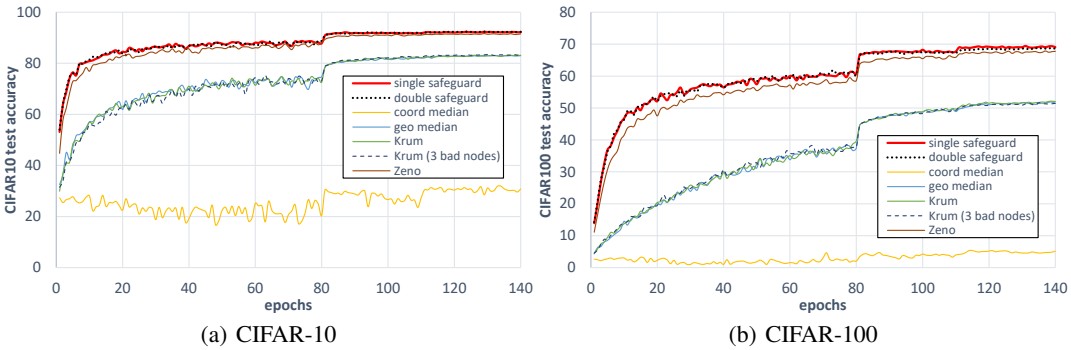

(a) CIFAR-10                    (b) CIFAR-100

Figure 6: Performance comparison under the label-flipping attack.

From the plots, one can see that our single and double safe-guard algorithms even outperform the "ideal accuracies." (92.4% accuracy vs "ideal accuracy" 91.7% under CIFAR-10; 69.4% accuracy vs "ideal accuracy" 68.0 under CIFAR-100.) In addition, we have found out that the safeguard algorithms *did not catch* any bad worker. This should not be surprising, since label-flipping (a.k.a. label smoothing) is known to be a regularization technique to actually improve test accuracy, as opposed to hurt performance.

Zeno also performs well under this attack (but it does not outperform the ideal accuracy). We have investigated into Zeno, and found out that it cannot distinguish good workers from bad workers under label-flipping attack; and therefore Zeno effectively always runs under 6 *random workers* as opposed to using the full power of the $m = 10$ workers (recall Zeno picks 6 workers with the topmost scores, see Definition C.4). This explains its (minor) under-performance comparing to safeguard.

Other prior works perform significantly worse, and this should be alarming since label-flipping is one type of smoothing technique to improve test accuracy, as opposed to an actual "attack" to hurt performance.

### C.2.5    SAFEGUARD ATTACKS

Finally, in the safeguard attack that we design, Byzantine workers send negative but re-scaled gradient to the master. We choose the re-scale factor so that it hardly triggers the safe-guard conditions at the master. From our experiment, choosing the re-scale factor as $0.6$ in all the cases do not trigger the safe-guard conditions, while choosing a re-scale factor as $0.7$ enables the algorithm to catch Byzantine workers occasionally. Our results are shown in Figure 7 (for re-scale factor $0.6$) and Figure 8 (for re-scale factor $0.7$).

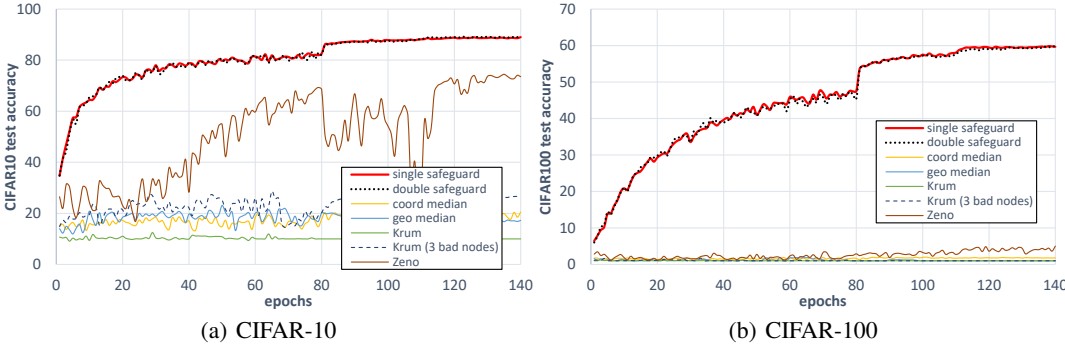

(a) CIFAR-10                    (b) CIFAR-100

Figure 7: Performance comparison under the safeguard attack with re-scale factor 0.6. (Recall this attack is designed to maximally impact the performance of our algorithm.)

**Re-scale factor 0.6.** In Figure 7, the performance of our (single and double) safeguard algorithms indeed get hurt a bit. Recall in Figure 7 the re-scale factor 0.6 is chosen to maximally impact our algorithm. The test accuracy drops from 91.7% to 89.3% under CIFAR-10; and drops from 68.0% to 60.0% under CIFAR-100 (for both single and double safeguards). In these cases, we confirm that both versions of the safeguard algorithms did not catch any bad worker. However, this still significantly outperforms all prior works.

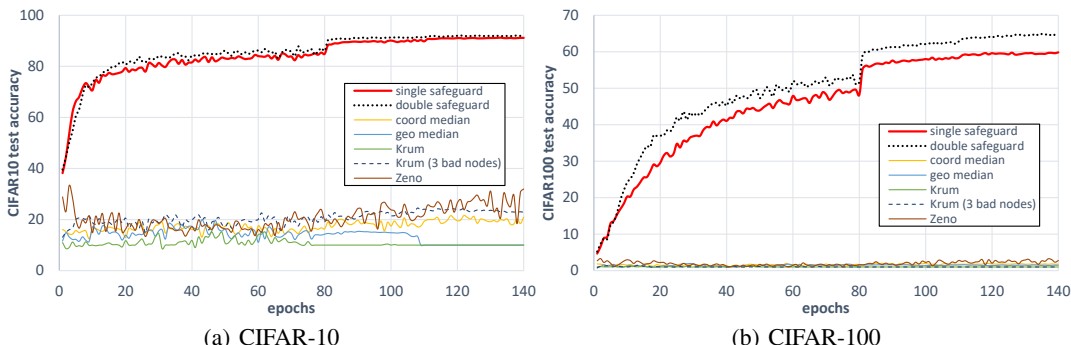

(a) CIFAR-10           (b) CIFAR-100

Figure 8: Performance comparison under the safeguard attack with re-scale factor 0.7. (In this case, our our algorithm can catch some bad workers, and thus perform nearly optimally.)

**Re-scale factor 0.7.** In Figure 8, we present the scenario when the re-scale factor is 0.7, so that the safeguard algorithms can *occasionally* catch some bad workers (depending on the randomness and learning rate). We confirm that in the three runs of *single* safeguard, it catches $1, 2, 3$ bad workers for CIFAR-10, and $1, 0, 0$ bad workers for CIFAR-100 respectively; in the three runs of *double* safeguard, it catches $1, 2, 4$ bad workers for CIFAR-10, and $2, 2, 2$ bad workers for CIFAR-100 respectively.

Since there is a significant performance gain when our safeguard algorithms catch bad workers, this explains why safeguard algorithms in Figure 8 outperform their counterparts in Figure 7 with rescale factor 0.6. At the same time, we notice that the double safeguard algorithm has the ability to *catch bad workers more easily*. This is why double safeguard significantly outperforms single sageguard in Figure 8.

In contrast, all other prior algorithms perform extremely bad under this attack. To some extent, safeguard attack is *even stronger* than the previously proposed *variance attack*, since it can drag the 100-class test accuracy on CIFAR-100 for all prior defense algorithms to nearly 1%, while variance attack can only drag them down to around 10%.

## C.3 FULL COMPARISON TABLE

We also include the full test accuracy comparison table in Table 1.

| | single safeguard | double safeguard | coord median | geo median | Krum | Krum (3 faulty nodes) | Zeno |
|---|---|---|---|---|---|---|---|
| *variance attack* | **92.02** | **91.75** | 21.43 | 22.01 | 21.47 | 22.4 | 33.42 |
| *sign-flipping attack* | **91.93** | **92.08** | 22 | 59.65 | 57.03 | 70.87 | 22.4 |
| *label-flipping attack* | **92.33** | **92.44** | 31.93 | 83.07 | 83.18 | 83.52 | **91.66** |
| *delayed-gradient attack* | **91.58** | **91.42** | 29.43 | 74.36 | 61.81 | 79.29 | 77.27 |
| *safeguard(x0.6) attack* | **89.01** | **89.26** | 21.44 | 23.12 | 12.48 | 28.66 | 74.46 |
| *safeguard(x0.7) attack* | **91.24** | **92.08** | 21.61 | 19.95 | 15.17 | 24.52 | 33.36 |

| | single safeguard | double safeguard | coord median | geo median | Krum | Krum (3 faulty nodes) | Zeno |
|---|---|---|---|---|---|---|---|
| *variance attack* | **68.27** | **67.95** | 6.6 | 5.81 | 5.05 | 6.1 | 10.87 |
| *sign-flipping attack* | **68.02** | **68.08** | 2.13 | 10.19 | 10.93 | 28.34 | 2.59 |
| *label-flipping attack* | **69.43** | **68.8** | 5.34 | 51.85 | 52.13 | 51.66 | **67.86** |
| *delayed-gradient attack* | **67.03** | **66.42** | 4.04 | 36.34 | 31.43 | 44.85 | 36.6 |
| *safeguard(x0.6) attack* | **59.87** | **60** | 2.01 | 1.9 | 1.25 | 1.72 | 5.02 |
| *safeguard(x0.7) attack* | 59.84 | **64.91** | 2.07 | 1.97 | 1.32 | 1.55 | 3.31 |

Table 1: Table of test accuracy performances for CIFAR-10 (above) and CIFAR-100 (below).

## C.4    ATTACK AGAINST THE CONVEX ALGORITHM OF ALISTARH ET AL. (2018)

We now briefly describe an attack against this algorithm. The attack specifically leverages the fact that the algorithm does not use sliding windows.

One can first run the vanilla SGD to compute "the maximum deviation per good worker" for the accumulation vector used by the algorithm $\sum_{t=0}^{T} \nabla_t$. This maximum deviation is therefore a lower bound for the threshold used in their algorithm. Next, we design an attacker who evenly distributes this total allowed deviation to e.g. 5 consecutive epochs, and behaves honestly for the remaining epochs. Such an attacker cannot be identified by this algorithm, because its total deviation across all the iterations is identical to that of a good worker. However, this leads the algorithm to divergence.

Specifically, suppose 4 Byzantine workers all maliciously report their stochastic gradients multiplied by the scalar $-5$, and the remaining 6 good workers report their true stochastic gradients. One can verify numerically that this attacker can run for 5 consecutive epochs (say, epochs $a, a + 1, a + 2, a + 3, a + 4$) without being caught by the algorithm. Now,

- if $a \leq 75$, within just 1 epoch of attack, the neural net weights diverge (value NaN).
- if $80 \leq a \leq 115$, this attack is applied after the first learning rate decay. Within just 1 epoch of the attack, the objective explodes and accuracy becomes $10\%$ (random), and within 3 epochs the algorithm diverges completely.
- if $120 \leq a \leq 155$, this attack is after the second learning rate decay. Within just 2 epochs of attack, the accuracy drops to $11\%$. Later, the accuracy never recovers above $40\%$.

