# OpenReview forum: "Byzantine-Resilient Non-Convex Stochastic Gradient Descent"
_ICLR.cc/2021/Conference — ICLR 2021 Poster_

### Official Review · AnonReviewer1 · 2020-10-26
**below acceptance threshold**

**Rating:** 5
**Confidence:** 4

**Review:**

In this paper, the authors propose a new algorithm called SafeguardSGD, which solves the Byzantine detection problem, and tolerate less than half Byzantine workers with theoretical guarantees. Both theoretical and empirical results are provided. Compare to the baselines, the proposed algorithms show better performance on fixed Byzantine workers. In overall, I think this is a good paper.

However, there are some issues to be resolved:

1. The problem solved in this paper is actually not Byzantine tolerance, but Byzantine worker detection, which is a special case of Byzantine tolerance with stronger assumptions. The main difference is that, in this paper, the proposed method assumes that the malicious workers never change their roles (once Byzantine, always Byzantine), i.e., the indices of the Byzantine workers never change. Such assumption is not required by most of the previous works (Krum, median, Zeno), although it may not be emphasized in the previous work. With such assumption, the proposed algorithm could make the use of the historical info and remove the Byzantine worker permanently, which is why it could perform better than the baselines. I'm ok with the scenario (which is also used in [1]), but the authors should clearly state that the threat model in this paper is weaker than the previous work.

Why this assumption matters?  Note that it is unnecessary that the Byzantine workers always behaves maliciously.

Considering the following cases:

(1) The Byzantine workers behave as honest workers at the beginning, and start the attack at the middle of the training. I believe the proposed algorithm could handle this case, but it will be better if such experiments could be added.

(2) On some workers, there are some occasional and temporal hardware/software failures that produce results with huge error (flipped bits, large random noise), but such workers will recover later. In this case, it seems unreasonable to remove the failed workers from the list of good workers permanently.


2. It seems that the proposed algorithm is highly related and similar to [1]. However, the difference to [1] is not discussed in details in this paper. Furthermore, since [1] uses the same weakened threat model as this paper, it should be the most important baseline to compare to, which is not included in the experiments. I understand that there is not experiments in [1], so doing experiments for it may be difficult, but I'm not sure whether it's just difficult or totally infeasible. At least, the authors should explain why [1] is not included in the experiments.

[1] Alistarh, D., Allen-Zhu, Z., & Li, J. Byzantine stochastic gradient descent. NeurIPS 2018.

---

> ### Author Response · Authors · 2020-11-17
> **Dear AnnoReviewer1: Thank you, we address your main questions below.**
>
> Question 1: Comparison to [1], the ByzantineSGD in the convex case? Why [1] is not included in the experiments?
>
> Short Answer 1: on the theory side, the analysis of this paper is completely different from [1]; on the practical side, algorithm [1] doesn’t work in the non-convex case at all (see below).
>
> Long answer 1: On the theoretical side, recall the same SGD algorithm admits completely different proofs in the convex vs non-convex cases. In the convex case, SGD convergence has a 3-line proof, but in the non-convex case, SGD requires long, delicate arguments to ensure approximate local minima, see e.g. 1902.04811 or the original 1503.02101. Thus, even the same SGD algorithm can require very new proofs in the nonconvex case, and can be considered as a significant contribution. We believe this makes our contribution relative to [1] very solid. Specifically, there is no proof overlap between our paper and [1]: e.g., there was no need to escape saddle points in a convex case.
>
> On the practical side, please note in [1], thanks to convexity, it suffices to check concentration of “sum_{t=0,1,..T} nabla_t”, without using sliding windows sum_{t=last, last+1,... T} like we do. This algorithmic difference means [1] could fail dramatically in the non-convex Byzantine case. A concrete experiment on CIFAR10 is given below.
>
> Experiment 1: consider same setting as our paper (resnet-20, 10 workers, 4 Byzantine workers, learning rate 0.1, decay at 80 and 120 epochs, etc). One can first run the vanilla SGD to compute “the maximum deviation per good worker” for the accumulation vector “sum_{t=0,1,..T} nabla_t”. This “maximum deviation” is therefore a lower bound for the threshold used in algorithm [1]. Next, we can design an attacker, who evenly distributes this total allowed deviation to 5 consecutive epochs, and behaves honestly for the remaining epochs. Such an attacker cannot be caught by [1], because its total deviation across all the iterations is identical to a good worker. However, it can totally break [1].
>
> Specifically, suppose 4 Byzantine workers all maliciously report “-5” times their stochastic gradients, and the remaining 6 good workers report their true stochastic gradients. One can verify numerically that this attacker can run for 5 consecutive epochs (say, epochs a, a+1, a+2, a+3, a+4) without being caught by algorithm [1]. Now,
> * if a<=75, within just 1 epoch of attack (not even needing 5 epochs), the neural net weights become NaN.
> * if 80<=a<=115, this attack is after the first learning rate decay. Within just 1 epoch of attack, the objective explodes and accuracy becomes 10%, and within 3 epochs it becomes NaN.
> * if 120<=a<=155, this attack is after the second learning rate decay. Within just 2 epochs of attack, the accuracy drops to 11%. Later, the accuracy cannot go above 40%.
>
>
> Q2: “The problem solved in this paper is not Byzantine tolerance but Byzantine worker detection ...… In this paper, the proposed method assumes that the malicious workers never change their roles (once Byzantine, always Byzantine)”
>
> A2: our threat model is the standard Byzantine fault model of distributed computing (originally introduced in [Lamport et al., ToPLAS 1982]). Specifically, the Byzantine model assumes that a fraction of the workers can behave arbitrarily, but that their IDs stay fixed.
>
> We stress that we do not assume that Byzantine workers have to always behave maliciously, or that our algorithm will always detect them. For instance, the Safeguard attack is designed so that our algorithm is never able to mark any of the workers as Byzantine, since they stay within the safeguard bounds. In such a case, our algorithm solves Byzantine-fault-tolerant SGD *without* Byzantine worker detection.
>
> Experimentally, we confirm that the Byzantine attackers can start to behave maliciously in the middle of the execution, as suggested, and the same performance will hold. (Please see the revision.)
>
> Question 3: (cont.) “such assumption is not required by most of the previous works (Krum, median, Zeno)...“
>
> Answer 3: Indeed, Krum, median, and Zeno filter out bad gradients in a round-by-round fashion, and so, the IDs of Byzantine nodes could change. However, the recent attack of [Baruch et al. “Circumventing defenses for distributed learning”, NeurIPS 2019] shows that such round-by-round schemes suffer from a serious vulnerability: an attacker can shift the true mean while being statistically indistinguishable from a correct worker, since it can always stay “within variance” at each round. Their paper breaks Krum and median in real-world scenarios. Thus, it is unclear whether any round-by-round filtering algorithm could defend against such an attack, without historical information or additional assumptions. Note that employing historical information implies that Byzantine workers cannot change IDs.
>
> By contrast, our Safeguard is able to defend against this SOTA attack, specifically due to leveraging historical information.

---

> > ### Comment · AnonReviewer4 · 2020-11-24
> > **While not a deal breaker for me, but Reviewer 1 has a solid point**
> >
> > I agree with Reviewer 1 that Byzantine failures that are temporally limited can arise due to non-malicious reasons. The proposed approach would fully eliminate such workers from any future consideration, which means one can never benefit from data stored at that worker. While this is not a deal breaker for me, I believe that Reviewer 1's comment should be carefully discussed in the paper. Whether the solution to this issue is round-by-round schemes or not is something that can be left for future work. However, bringing this issue out in the paper is important before the revision deadline closes.

---

### Official Review · AnonReviewer3 · 2020-10-28
**The paper has great formal analysis but is needs format organization and clarification**

**Rating:** 6
**Confidence:** 2

**Review:**


The authors present quite an interesting approach where they focus on the number of stochastic gradient evaluations rather than bounding the number of sampled stochastic functions. They converted other aggregation functions in their own formulation and then did a fair comparison. The results are in favor of the authors' method. The SafegaurdSGD algorithm is presented with several formal analysis and authors shows that why it is theoretically better to have two matrices (history record or safeguards) to mark the working machines good or bad/byzantine. The time complexity they present seems a bit difficult to understand and the argument they present for parallel speedup is also difficult to grasp as the machines are already running in a distributed environment how one can do parallel speedup? This should be presented in a clear way. The formal analysis is though rigorous but it requires a lot of time to understand as some of these notations are not defined and clearly mentioned and one has to assume some of the arguments presented to follow on.  The author presents that their method offers stronger bounds.  I appreciate the formal work and it clearly aligns with the results they have shown in the paper and in the supplementary part. The authors have studied the latest attacks introduced in literature and experimented other methods in their own introduced attack for a fair comparison which is great. Although there are several parameters that the authors do not discuss how they are chosen for the experiments they just show the experiment results for some selected values of parameters.
The paper needs some format organization and need to define and clear something so it might become clear to other readers to understand and grasp concepts.
The authors have already talked about the parallel speed in the introduction and instead of iterating in after Theorem 2.3, space can be used to fill with other detail or definition as it can be seen the authors are already short on space due to page limit.
In the end, I want to say that please define notations before using them. It is quite an issue while reading the paper, you have to stop and have to find the definition. $\Omega(m^2d)$ in the introduction $d$ is not defined before. While mentioning the number of iteration required $x_0$ is not defined, $\Tilde O$ is not defined and is defined at section $2$. Lemma 3.2 $\Delta_t$ is used and then defined later. The spelling of Loewner is wrong.

---

> ### Author Response · Authors · 2020-11-17
> **Dear AnnoReviewer3:**
>
> Thank you for the comments; below we summarize your major concerns and address them one by one.
>
>
> Q1: the machines are already running in a distributed environment, how can one do parallel speedup?
>
> A1: We mean the total distributed running time, compared to running everything on a single (honest) machine. Recall if there are no Byzantine workers, the (wall-clock) speed-up factor is m for SGD (when ignoring communication overhead). We thus call it “parallel speedup” following some tradition. We will explain this better in our revision.
>
> Q2: Please define notations before using them. It is quite an issue while reading the paper, you have to stop and have to find the definition. Omega(m^2 d) in the introduction is not defined before. While mentioning the number of iteration required x0 is not defined, \tilde{O} is not defined and is defined at section 2. Lemma 3.2 Delta_t is used and then defined later.
>
> A2: We apologise, we will fix these issues in the revision.
>
> Q3: The formal analysis is though rigorous but it requires a lot of time to understand as some of these notations are not defined and clearly mentioned and one has to assume some of the arguments presented to follow on.
>
> A3: Thank you, we acknowledge your comment and will strive to improve in the revision. Meanwhile, we would like to mention that we have put significant effort to simplify analysis and notations. If you go to page 24-29 of the original paper by Jin et al (2019), see arXiv 1902.04811, you may notice that we have really tried our best to simplify notations, so that our Byzantine version of the proof is even shorter than the original Jin et al (2019), which has no Byzantine workers!
>
> Some of our simplifications come at the expense of adding a few minor assumptions (such as replacing E[| ..|^2]<=sigma^2 with an absolute bound |..|<=sigma. This has already made some reviewers (e.g. AnnoReviewer2) question why we do so. We have tried to find a delicate balance between what assumptions we make, and how simple the proof can be. We understand we can do even better, and will follow your suggestions.

---

### Official Review · AnonReviewer4 · 2020-11-03
**A good paper on Byzantine-resilient distributed optimization; does need additional clarifications and discussion that *might* be doable in a revision.**

**Rating:** 7
**Confidence:** 3

**Review:**

This paper studies distributed non-convex learning that is Byzantine resilient. The main contribution of the paper is an algorithm that is based on perturbed stochastic gradient descent, that is provably resilient to Byzantine failures, and that is guaranteed to reach near second-order stationary points of non-convex objective functions. Compared to similar works in the literature, this paper focuses on identifying Byzantine workers using a "concentration of gradients" argument that are then permanently removed from consideration for future iterations. The paper has a nice blend of algorithmic development, theoretical analysis, and detailed experiments. Overall, it is a good paper, worthy of publication in the proceedings. Before the final decision, however, I would like the authors to help address the following questions / comments of mine.

**Major Comments**

1. The permanent removal of workers from each iteration appears to be a risky strategy when one is dealing with workers' data that are not i.i.d. In particular, "RSA: Byzantine-robust stochastic aggregation methods for distributed learning from heterogeneous datasets" is one such work that seems to be able to deal with non-i.i.d. datasets in Byzantine settings. It would be useful to see how does the proposed approach work in the presence of non-i.i.d. datasets and how does it compare to works such as the one referenced earlier.
   - A minor point, which is more of a curiosity: Is it possible for the Byzantine workers to play a colluding game where they force convergence **before** the algorithm had a chance to go through the two windows and eliminate the Byzantine workers?
2. While the paper refers to its strategy as "stochastic gradient descent", the method is really "perturbed stochastic gradient descent" that involves adding white Gaussian noise of variance $\nu^2$ to the iterates. This adds one more parameter to the problem and it is not clear how would one tune this parameter. Theorem 2.3 in the paper requires $\nu^2$ to be set according to knowledge of $\alpha$, the fraction of Byzantine workers. It is not clear from reading the paper as to how the authors set this parameter in their experiments.
3. Related to the above point, Theorem 2.3 only provides scaling guidelines for the different parameters (including the stepsize) in SafeguardSGD and even those scaling guidelines involve knowledge of $\alpha$. How does one fix these parameters and how were these parameters set for the experiments? In particular, one wonders why does the stepsize need to be a function of the fraction of Byzantine workers?
4. The statement of Theorem 2.3 needs a bit of clarification. First, it would be useful to specify the "high probability" part. What's the scaling of this high probability and it scales with which parameters? Second, what is the practical significance of "for at least constant fraction of the indices $t$"? Does this mean that the algorithm can, at the final stages of the algorithm, actually leave the neighborhood of the stationary point?
5. In Theorem C.1, part (c), the probability is lower bounded by 0.45. This does not seem like "high probability" at all. What are the implications of this assumption in Theorem C.1, part (c) on the overall results reported in the paper?
6. The experimental results are done for fixed numbers of Byzantine workers. It would have been useful to see how the performance scales with the increase in the number of Byzantine workers.
7. While this paper discusses distributed learning, Byzantine resilience in decentralized learning has also been investigated in recent works (see, e.g., Adversary-resilient distributed and decentralized statistical inference and machine learning: An overview of recent advances under the Byzantine threat model). Are there any lessons for distributed Byzantine resilience from this paper that can be translated to decentralized Byzantine resilience?

**Minor Comments**

1. The range of $p \in (0,1)$ should be specified in Lemma 3.1 for complete rigor.

***Post-discussion period comments***

I am pleased with the revision the authors have posted during the discussion phase. I am also satisfied with the authors' response to my comments and to the comments of the other reviewers. I therefore recommend that this paper be accepted into proceedings of ICLR 2021.

---

> ### Author Response · Authors · 2020-11-17
> **Dear AnnoReviewer4:**
>
> Thanks for careful reading of our paper. Below we only address your major comments in order (to save space).
>
> Q1: How about non-i.i.d. (heterogeneous) data?
>
> A1: Similar to some prior work, we study the case when workers have i.i.d. data. This is realistic since in many neural net training, e.g. when training ImageNet, the entire training data is usually shared across workers. We admit it can be challenging to study non-i.i.d. data for the nonconvex objectives, so this is left as future work.
>
> Q2: Should be “perturbed” SGD instead of SGD.
>
> A2: Fair point. We have followed tradition (since the seminal work 1503.02101 and others) to call “perturbed SGD” also SGD for short, because in practice, the random noise of SGD is sufficient to help escape saddle points, and the random Gaussian is added only for theoretical purposes.  Adding Gaussian perturbation in practice makes no difference (both for our alg and classical SGD).
>
> Q3: Theorem 2.3 only provides scaling guidelines for the different parameters ... those scaling guidelines involve knowledge of alpha. How were these parameters set for the experiments?
>
> A3: Such scaling guidelines are also common in prior works (see e.g. arXiv 1803.08917). One can derive different performance gains for different ranges of parameters (see for instance Theorem 3.8 of 1803.08917), and we have simply picked the “optimal parameters” in our theorem statement for a fair theoretical comparison.
>
> In practice, there is no strict requirement, and we found a wide range of parameters to work. Our experiment parameters are given on Page 19. For instance, the \nu parameter doesn’t matter (see our answer A2 above) and setting \nu=0 also works. The safeguard thresholds can be automatically derived either by pre-running the algorithm or calculating them on the fly. We chose the window sizes T1 and T2 without tuning. We set the learning rate to be 0.1, which is standard for this model/dataset in the failure-free case.
>
> Q4: It would be useful to specify the "high probability" part. What's the scaling of this high probability and it scales with which parameters? Second, what is the practical significance of "for at least constant fraction of the indices t"?
>
> A4: High probability means that if we want probability 1-p, then the final bounds only have log(1/p) factors. We hide it to make the theorem statement clean. In fact, we have carried around these log parameters in our proofs. We only omitted them in the last step, to keep the statement clean.
>
> As regarding “at least a constant fraction”, this is necessary for theoretical purposes. All existing non-convex convergence results require it. See for instance the very first paper by Nesterov 2008 (Thm 5 in 1902.04811). Furthermore, Thms 10,13,16,17 in 1902.04811 even stated a weaker version “meet a stationary point at least once”, as opposed to “at least constant fraction of the iterations”. Imagine, one can construct a nonconvex function where if you stop an algorithm at a fixed iteration T, it is on the way escaping one local minima and moving to a better local minima. This is why we cannot guarantee for all the indices in theory. This will not happen in the convex case, and not likely happen in practice in the nonconvex case.
>
> Q5: In Thm C.1, probability is lower bounded by 0.45. This does not seem like "high probability"
>
> A5: There is no contradiction. Thm 2.3 states that, w.h.p., for at least a CONSTANT fraction of the iterations, the result holds. This constant fraction can be for instance 30% of the iterations. As we said in Answer A4, this is *necessary* for non-convex theory and prior work (e.g. 1902.04811) often guarantees it for at least ONE iteration. We improved this to 30% of the iterations; but this cannot be improved to the last iteration only.
>
> Q6: experiment result for increasing the number of Byzantine workers?
>
> A6: Thanks. We currently used m=10 workers with 4 Byzantine workers (the hardest possible), and batch size 8 (so 80 samples per iteration). This total batch size 80 is a good choice for training residual models on CIFAR10. We have also tried 100 workers with 40 Byzantine workers and batch size 1 and there’s no significant difference --- because the total number of samples per iteration remains the same. We could run for even more samples, but then run into batch scaling issues.
>
> Q7: Are there any lessons for distributed Byzantine resilience from this paper that can be translated to decentralized Byzantine resilience?
>
> A7: We believe additional assumptions are needed. If the Byzantine workers form a node cut to block honest workers from communication, then there’s no solution. If the network forms a tree, we can ask the machines to pass gradients to the root; then, each non-Byzantine machine can keep a safeguard (like this paper) for its subtree. We believe this is a very interesting but challenging question, and we leave it a future work to study what’s the reasonable assumption to make.

---

### Official Review · AnonReviewer2 · 2020-11-04
**Good paper, accept**

**Rating:** 8
**Confidence:** 4

**Review:**

The paper considers stochastic gradient descent convergence in a distributed setting with m workers, where up to α workers can be Byzantine, i.e. perform in an arbitrarily adversarial way. In this setting, they develop a variant of SGD which finds a second-order stationary point, prevents Byzantine workers from significantly affecting convergence, and achieves α^2 + 1/m speedup compared with the sequential case. The main idea of the algorithm is to measure deviations of gradient updates for a certain number of rounds and detect Byzantine machines which must have a significant deviation to noticeably affect the algorithm’s behavior.


If I’m correct, Lemma 3.1 allows a much simpler proof:
|\sum_{t=1}^{T-1} (ξ_0 + … + ξ_{t-1}) * Delta_t|
<= |\sum_{t=1}^{T-1} ξ_t| * |\sum_{t=1}^{T-1} Delta_t| + |\sum_{t=1}^{T-1} ξ_t * (Delta_1 + … + Delta_t)|
The first term is estimated as a sum of independent Guassians. The second term can be estimated using Azuma’s inequality (it’s a martingale since E[ξ_t] = 0 and ξ_t is independent on ξ_1, …, ξ_{t-1}, Delta_1, …, Delta_t). We can bound their norms of ξ_t by O(log (T/p)) with probability 1 - p/T.

Questions:
In Jin et al. (2019), dependence on d can be avoided when an additional assumption (Lipschitz stochastic gradient) is made. Can this work use this assumption? I believe that footnote 3 on page 4 talks about this assumption and argues that it may be too strong for practical applications, but I don’t see a reason to not get a result with this assumption. Are there technical obstacles?
Assumption 2.1, both items: the bounds are typically taken in expectation: E[||∇f(x_t) - ∇_ti ||^2] <= σ^2. Can this paper handle this kind of assumption? At the very least, one can’t immediately detect byzantine machines that deviate from the mean by more than σ.
Algorithm 1, line 11: it shouldn’t matter, but isn’t it more natural to take an average over good_{t+1}, not good_t?
Page 13, last equation: while the inequality seems to be true (since ξ_t are independent Gaussians), I don’t see how it follows from Lemma 4.2.
Page 14, Equation B.2: isn’t this way to bound the sum too loose? What’s the intuition behind this approach? Can we get a better bound with some other approach?

The following parts would benefit from an additional discussion:
While it’s clear that α^2 + 1/m comes from bounds on σ_t and Delta_t, the intuitive meaning behind the α^2 term is not clear. The reason why α=1/sqrt(m) is a threshold is also not clear.
While it may be clear from the algorithm (namely how the median behaves), I think it’s also worth explaining why the algorithm doesn’t significantly degrade when α is close to ½, unlike what one can expect.

I believe that the following places would benefit from the further expansion:
Page 13, bound on ||Delta_0 + … + Delta||^2: it’s better to explicitly write relation between B^t and Delta_t.
Page 15, “for some vector ||θ_t||”: I would explain the bound.
Page 15, “ψ_t is zero except in the first coordinate”: I would explain why.
Page 16, right after M is introduced: I would explain the first transition.

Minor issues:
While assuming that smoothness constants are 1 slightly simplifies the presentation, I believe that it makes some transitions harder to understand. E.g. bound on θ_t on page 15 would be more clear if the Hessian Lipshitz constant was in the equation.
nu should be an input/parameter of Algorithm 1
Page 13, proof of Claim B.2: it’s said that the proof is by induction, by I don’t think you use induction anywhere.
Page 13, Proof of Lemma 4.3: “Lipschitz smoothness” -> “smoothness”?

---

> ### Author Response · Authors · 2020-11-17
> **Dear AnnoReviewer2:**
>
> Thanks for your time and support. Below we try to address all of your main concerns.
>
> Q1: In Jin et al. (2019), dependence on d can be avoided when an additional assumption (Lipschitz stochastic gradient) is made. Can this work use this assumption? Footnote 3 on page 4 argues that it may be too strong for practical applications.
>
> A1: Thanks for asking.
>
> Short answer: (1) yes, we believe this assumption (which we didn’t use) is “provably” unrealistic, and (2) yes, our analysis can be adapted to leverage this assumption, but it requires an algorithmic change that we do not think is worth adding.
>
> Longer answer: first, this assumption is indeed too strong, at least for neural net training. In particular, an individual sample’s Lip-smoothness can be 100++ times worse than the overall Lip-smoothness, even at the end of the training. One can either verify this by experiment (in which we did on CIFAR data), or compare this to existing work on adversarial examples: it is known that even an extremely small perturbation can drastically change the output (thus the cross-entropy gradient) for an *individual* sample on a clean-trained model.
>
> Second, algorithms based on this assumption are often impractical for neural net training. For instance, variance reduction (VR) algorithms require this additional assumption to work; although beautiful in theory --- with 1/eps^3 rate at best, see arxiv 1807.01695 --- VR-based algs unfortunately do not beat SGD in practice.
>
> Third, if this assumption is made, there are two ways to modify our paper. The first one is to Byzantine “robustify” a VR-based alg (this is possible!) to get a 1/eps^3 rate. We don’t think it’s worth doing it, since VR-based algs do not beat SGD in neural net training. The other way is to derive the 1/eps^4 rate for SGD, but this requires an algorithmic change --- namely, to add a third safeguard to ensure “the stochastic nabla_t - nabla_0 is small and proportional to |xt-x0|” under a certain window size. Again, since this assumption is not realistic, we do not wish to influence the readers to add this algorithmic change. But, it is indeed possible.
>
> Q2: Assumption 2.1, the bounds are typically in expectation: E[||∇f(x_t) - ∇ti ||^2] <= σ^2. Can this paper handle this assumption?
>
> A2: Yes we can, at the expense of complicating the proofs and introducing additional log factors. For instance, we used Lemma B.1 (Pinelis 1994) to derive concentration using absolute bounds, and if that changes to E[|..|^2] then one needs to complicate the notations and derive some vector version of Azuma inequality. We wished to make this paper simple enough comparing to e.g. Jin et al (2019) to attract more readers. Thus, we used this assumption.
>
>
> Q3: Algorithm 1, isn’t it more natural to take an average over good{t+1}, not good_t?
>
> A3: You are right. We chose the latter since it makes our proof simpler. This way, the denominator no longer depends on the randomness/adversarial choice at the current round. Of course, one can switch it to good_{t+1} in the end and pay a negligible additive term.
>
> Q4: Page 13, last equation: I don’t see how it follows from Lemma 4.2.
>
> A4: Thanks! That’s a typo. Ignoring C1 for notational simplicity, Lemma 4.2 says <nabla f(wt), xi_t> <= |nabla f(wt)| * O(eta v) <= 0.05 eta |nabla f(wt)| + O(eta v^2). Then sum it up over t.
>
>
> Q5: Page 14, Equation B.2: isn’t this way to bound the sum too loose? What’s the intuition behind this approach? Can we get a better bound?
>
> A5: Short answer: you’re absolutely right, that individual bound is loose. However, since we are not losing significant factors (once C2 is replaced with 1) compared to Jin et al. (2019), our analysis is still asymptotically tight.
>
> Longer answer: if one doesn’t want to use Equation B.2, then our best guess is that one needs to add an additional safeguard to ensure “<nabla f(w_t), nabla_t>” also concentrates. This is different from the simple stochastic gradient concentration, so we want to keep the algorithm as simple as possible.
>
>
> Q6: suggest adding discussions for why the algorithm doesn’t significantly degrade when α is close to ½, unlike what one can expect.
>
> A6: Thanks for this great suggestion. Short answer, with m machines the variance of the stochastic gradient usually degrades to 1/m of it used to be. However, with alpha*m machines colluding, they can contribute to this variance by alpha^2 additively. This gives the (alpha^2 + 1/m) that shows up in prior work and in this work.
>
>
> Q7: simple proof of Lemma 3.1
>
> A7: thank you and we believe you’re right. Our main difficulties in the proofs are regarding how to use the “minimal safeguard” to derive tight bounds for SGD in the non-convex setting, and we believe this requires us to have quite different proofs from Jin et al. (2019).
>
> Q8: I believe that the following places would benefit from the further expansion. [...]
>
> A8: Thanks a lot! The above are all good suggestions, which we will implement in the next revision.

---

### Decision · Program_Chairs · 2021-01-07
**Final Decision**

**Decision:**

Accept (Poster)

**Comment:**

The paper presents a new algorithm for byzantine resilient nonconvex distributed optimization. The presentation is clear, the motivation is solid, and the problem setting is interesting. The novelty of the present work is sufficient for publicaiton. The new scheme comes with some provable guarantees, improving the prior state of the art. Some of these guarantees are arguably not corresponding to strong operational robustness guarantees, however they compare well with convergence proofs of the related literature. Some concerns were raised with regards to comparison with some prior work, but the authors addressed it in the rebuttal.